# Information Bargaining: Bilateral Commitment in Bayesian Persuasion

## Abstract

Bayesian persuasion studies how an informed sender can influence a receiver's actions through committed signaling schemes. While effective in one-shot settings, extending Bayesian persuasion to real-world long-term interactions becomes NP-hard. A separate empirical mismatch is also evident, where people tend to be much more truthful in practice than the signaling scheme predicted by Bayesian persuasion equilibria. To address these issues, we first prove that long-term Bayesian persuasion can be decomposed into a bargaining stage and a realization stage while preserving optimality and equilibria. This decomposition disentangles the previously conflated informational and first-mover advantages of the sender. Based on the results, we establish a unified, realistic, and fairness-oriented framework called *information bargaining*. Variants of bargaining-game settings and their associated solution concepts, such as the Nash cooperative bargaining model and the Nash bargaining solution, can be applied directly to persuasion games to yield more realistic models. To validate our framework, we identify and employ capable reasoning LLMs that can solve persuasion game equilibria effectively. In long-term persuasion task variants and in the corresponding bargaining game variants, these capable LLMs demonstrate that they reach the same equilibrium.

## 1 Introduction

Communication plays a pivotal role in human society, shaping interactions and influencing decision-making processes. The concept of "cheap talk", introduced by Crawford & Sobel (1982), underscores the importance of verbal communication in strategic contexts. McCloskey's assertion that "one quarter of GDP is persuasion" illustrates how integral persuasion is to economic activities (McCloskey & Klamer, 1995). Recent analyses by Antioch (2013) suggest that this figure has risen to 30%, highlighting the growing significance of persuasive communication in the modern economy. In this landscape, the idea of Bayesian persuasion, proposed by Kamenica & Gentzkow (2011), emerges as a crucial framework, involving the transmission of signals to influence a receiver's beliefs and actions. This framework finds a wide range of applications across various domains, including routing systems (Das et al., 2017; Kremer et al., 2014), law enforcement deployment (Hernández & Neeman, 2022; Lazear, 2006), and grading in schools (Boleslavsky & Cotton, 2015; Ostrovsky & Schwarz, 2010), as surveyed by Kamenica (2019).

A series of studies through the lens of Bayesian Correlated Equilibrium (BCE) demonstrates the high relevance of Bayesian persuasion. Following Bergemann & Morris (2013; 2016), Bayesian persuasion can be understood as the design of an optimal Bayes-correlated equilibrium: the sender **commits** *ex ante* to a signaling scheme that the receiver will find incentive-compatible to follow (Bergemann & Morris, 2019). As noted by Dughmi (2017), the commitment assumption is essential to Bayesian persuasion, otherwise the model will degenerate to the cheap talk model (Crawford & Sobel, 1982). Compared to the analysis of partition equilibrium in cheap talk, which suggests that cooperation becomes impossible when the conflict of interest is too large, the introduction of commitment enables concavification (Kamenica & Gentzkow, 2011) and Bayes Correlated Equilibrium (BCE) analysis to support deeper cooperation in a broader range of scenarios.

While effective in one-shot scenarios, Bayesian persuasion faces significant complexity (NP-hardness) in long-term interactions, where receivers may adopt dynamic strategies conditional on past outcomes and future expectations (Gan et al., 2022), prompting various online strategic propos-

als. For example, Gan et al. (2022) developed threat-based meta signaling schemes, and Bernasconi et al. (2024) explored promise-form signaling schemes, prompting various strategic proposals. Despite these advances, the literature currently lacks a unified theoretical framework capable of systematically integrating these diverse strategies. Practical experience in everyday life indicates a further discrepancy: people tend to behave more truthfully in practice than the exploitative and deceptive signaling scheme predicted by Bayesian persuasion equilibria. This underscores a persistent gap in current persuasion models' ability to account for realistic scenarios.

To address these gaps, we introduce the bargaining perspective to Bayesian persuasion. This perspective reframes classic one-sided persuasion as a balanced information bargaining framework by explicitly acknowledging the common knowledge of the game structure and providing the receiver comparable commitment capabilities. With only modest modifications to standard assumptions, we develop an information-bargaining framework for long-term Bayesian persuasion that delivers three contributions: **(1)** We show that long-term persuasion problems can be decomposed into a bargaining stage and a realization stage without affecting optimality or equilibria. This separation isolates the strategic negotiation of signaling scheme from their subsequent implementation and observation. **(2)** We clarify two advantages that have been conflated in Bayesian persuasion, namely the sender's informational advantage and the first-mover advantage. This clarification of previously conflated advantages, together with the decomposition of interaction stages, suggests that the receiver can be endowed with bargaining power. **(3)** Based on these structural insights, we establish a unified, realistic framework in which variants of bargaining game settings and their corresponding solution concepts can be directly applied to persuasion games. This mapping brings well-structured tools for long-term persuasion and delivers desirable properties such as fairness and Pareto efficiency, for example as captured by Nash bargaining solutions (Nash et al., 1950).

To empirically evaluate our information bargaining framework, we leverage recent advances in LLMs, which have demonstrated substantial improvements in reasoning capabilities (Guo et al., 2025). These models are increasingly employed as solvers for complex strategic problems. In our study, we use LLMs as equilibrium solvers to directly test our information-bargaining framework. We choose them specifically for their ease of implementation and transparent chain-of-thought, which facilitates rapid identification of cognitive hierarchy levels. Our proposed information-bargaining framework is validated through a two-stage pipeline by LLMs: (1) LLM game-solving accuracy evaluation. We begin by evaluating several LLMs on tasks with known theoretical results to assess their capacity as game solvers. Among them, `GPT-o3` (OpenAI, 2025) and `DeepSeek-R1` (Guo et al., 2025) pass our validation criteria. (2) Hypothesis validation. We subsequently apply these capable LLMs to compute equilibria within our bargaining framework, with results supporting our claims.

To make the manuscript broadly accessible for people who do not specifically work on information design and Bayesian persuasion, we have included in **Appendix A accessible explanations** to explain in plain language the content of each section and the contributions of our framework.

## 2 PRELIMINARIES

To be self-contained, we briefly introduce the frameworks of cheap talk, Bayesian persuasion, and bargaining in this section. All notations are summarized in the glossary (Appendix B), and the full procedures of all games are included in Appendix C to facilitate cross-game comparison.

### 2.1 CHEAP TALK AND BAYESIAN PERSUASION

Cheap talk models cost-free, non-binding messages between two players (Crawford & Sobel, 1982; Farrell & Rabin, 1996). Procedure 5 in Appendix C (identical to Procedure 1 but without the blue commitment steps) formalizes the game. In perfect Bayesian Nash equilibrium, such talk can sustain cooperation when goals are not too misaligned (Crawford & Sobel, 1982; Crawford, 1998).

Bayesian persuasion introduces a **commitment assumption**: the sender commits to and then follows a signaling scheme chosen before the state is realized (Kamenica & Gentzkow, 2011; Kamenica, 2019). Commitment may arise from repeated-game reputation (Dughmi & Xu, 2016; Dughmi, 2017), institutional authority (e.g., grading policies (Dughmi & Xu, 2016) or legislation (Kamenica, 2019)). This additional lever enlarges the set of cooperative equilibria relative to cheap talk (Ka-

menica & Gentzkow, 2011). The procedure of Bayesian persuasion is described as Procedure 1, where a sender $i$ is trying to send signals $\sigma \in \Sigma$ to influence the actions $a \in \mathbb{A}$ of a receiver $j$. (1) A neutral environment will first samples a state $s \in \mathbb{S}$ according to a prior distribution $\mu_0 \in \Delta(\mathbb{S})$. State $s$ is observable only for the sender, and it influence both players' payoffs. So the receiver has interests in state $s$ without knowing it, which gives an opportunity to the sender to influence the behavior of the receiver. This refers to the sender's **informational advantage.** (2) Next, the sender sends a signal $\sigma \in \Sigma$ to the receiver. Its signaling scheme is defined as $\varphi : \mathbb{S} \to \Delta(\Sigma)$, and the signaling scheme set is $\Psi$. (3) Then, the receiver takes an action $a \in \mathbb{A}$ given the signal. Its action rule is defined as $\pi : \Sigma \to \mathbb{A}$, and the action rule set is $\Pi$. The receiver's behavior is default to be Bayesian, which gives the task name "Bayesian persuasion". In the cheap talk setting, the receiver behaves in the same way: after receiving a signal, it updates its posterior belief and chooses the optimal strategy based on that belief. (4) The payoff functions of the sender and receiver are defined as $r^i : \mathbb{S} \times \mathbb{A} \to \mathbb{R}$ and $r^j : \mathbb{S} \times \mathbb{A} \to \mathbb{R}$, respectively. They do not generally share the same payoff function, so the task is mixed-motive at the most time. Any payoff instance is independent to the sender's strategy value $\sigma$, so for the sender to get better payoffs, it must leverage its informational advantage to manipulate the receiver to act towards its (the sender's) favor.

---

**Procedure 1:** Timing of Bayesian Persuasion

**Input:** Game settings $(\mu_0, \mathbb{S}, \Sigma, \mathbb{A}, r^i, r^j)$
1 *The sender decides a signaling scheme $\varphi$ ;*                    // Commitment (1)
2 *The sender commits $\varphi$ to the receiver ;*                    // Commitment (2)
3 Environment samples a state $s \sim \mu_0(\cdot)$ ;
4 The sender signals $\sigma \sim \varphi(\cdot \mid s)$ *as committed* ;                    // Commitment (3)
5 The receiver takes an action $a \sim \pi(\cdot \mid \sigma)$ ;
6 The sender and the receiver get rewards of $r^i(s, a)$ and $r^j(s, a)$ respectively ;

---

As stated, the receiver needs to make a Bayesian best response, which requires knowing the sender's signaling scheme $\varphi$. This is allowed in Bayesian persuasion and is referred to as the commitment assumption. In cheap talk, the receiver does not know the sender's signaling scheme in advance; it is only used in the analysis of equilibrium.

**Assumption 2.1** (Commitment). *The sender commits to a signaling scheme $\varphi$, which encompasses the following three key aspects: (1) The sender will decide on a $\varphi$ before the game starts, (2) The sender will honestly inform the receiver of this $\varphi$, and (3) during the game, the sender will actually samples signals $\sigma \in \Sigma$ according to the committed $\varphi$.*

We can further simplify the problem without loss of generality by assuming that the sender "sending a signal" is equivalent to "recommending an action" for the receiver to take. And this is known as an analysis similar to the revelation principle, proposed and proved in Kamenica & Gentzkow (2011).

**Proposition 2.2** (A Variant of Revelation Principle, Paraphrased from Dughmi (2017)). *Assuming that the signaling set is equal to the action set ($\Sigma = \mathbb{A}$) does not affect the optimality of the sender's signaling scheme $\varphi$. This implies that every signaling scheme is equivalent to one that recommends actions.* [1]

Let $\mathbb{V}^i$ and $\mathbb{V}^j$ denote the visibility set of the sender and receiver, respectively. Let $\mathbb{V}^{i+j}$ denote the common knowledge set (known by the players, with each aware that others know it, ad infinitum). Then $\mathbb{V}^i \setminus \mathbb{V}^j = \{s\}$ and $i, j, \mu_0, \mathbb{S}, \mathbb{A}, r^j, \varphi, \sigma, a \in \mathbb{V}^{i+j}$. The visibility of $r^i$ is not specified in previous work. Here we assume $r^i \in \mathbb{V}^{i+j}$.

To further characterize the task, we can restrict our attention without loss of generality to a subset of $\Phi$ whose elements are *signaling schemes that the receiver would respect*. This can be clearly introduced by the concept of **incentive compatibility**.

**Definition 2.3** (Incentive Compatibility in Bayesian Persuasion, Single-Receiver Version without Types of Bergemann & Morris (2016); Dughmi (2017)). *The sender's recommendation $\varphi$ is incen-*

---

[1] We will refer to this as the revelation principle in the following content since there will be no ambiguity here. We keep using the notation $\sigma$ to denote an action sent by the sender, using $a$ to denote an action actually taken by the receiver.

*tive compatible for the receiver if for each $a \in \mathbb{A}$,*

$$\sum_s \mu_0(s) \cdot \varphi(\sigma = \sigma \mid s) \cdot r^j(s, a) \geq \sum_s \mu_0(s) \cdot \varphi(\sigma \mid s) \cdot r^j(s, a'), \tag{1}$$

*for all $a' \in \mathbb{A}$.*

Every signal (recommended action) will induce a posterior belief of the receiver, who will then choose the Bayesian best response to it. If the receiver's best response is exactly the action recommended by the sender, then the receiver is considered *persuaded* by the sender. In this case, the receiver has no incentives to deviate from the sender's recommendation, i.e., $\pi(a \mid \sigma = a) = 1$, and thus can be omitted in this canonical setting. In the origin paper Bergemann & Morris (2016), it is named as the "obedience". **Bayes correlated equilibria (BCE)** is defined as the set of obedient signaling schemes.

Having these, the sender faces an optimization problem in the beginning of the game (Line 1 in Procedure 1):

$$\max_{\varphi} \quad \mathbb{E}_{\mu_0, \varphi, \pi}\left[r^i(s, a)\right] := \sum_s \mu_0(s) \sum_a \varphi(\sigma = a \mid s) \cdot r^i(s, a)$$

$$\text{s.t.} \quad \sum_s \mu_0(s) \cdot \varphi(\sigma = a \mid s) \cdot \left[r^j(s, a) - r^j(s, a')\right] \geq 0, \quad \forall a, a' \in \mathbb{A}. \tag{2}$$

That is, the sender is to select an incentive compatible signaling scheme for the receiver, to maximize its (the sender's) own expected payoffs. We refer to the canonical Bayesian persuasion as the formulation in Dughmi (2017). Concrete examples of persuasion are provided in Section 5.1.

**Definition 2.4** (Bayesian Persuasion). *Under Assumption 2.1 and Proposition 2.2, a Bayesian persuasion task is defined as $\mathcal{BP} := (\mathbb{S}, \mu_0, \mathbb{A}, r^i, r^j)$, where a sender tries to persuade a receiver according to Procedure 1.*

### 2.2 Bargaining Game

The bargaining game, initially introduced by Nash et al. (1950); Nash (1953), serves as a foundational model in the study of strategic negotiation. We first introduce the definition in Maschler et al. (2013).

**Definition 2.5** (Cooperative Bargaining Game (Maschler et al., 2013)). *A two-player bargaining game is defined as $\mathcal{BG} := (\mathbb{Y}, \boldsymbol{d})$, where $\mathbb{Y} \subseteq \mathbb{R}^2$ is a nonempty, compact **feasibility set** of possible agreements, and $\boldsymbol{d} = (d_i, d_j) \in \mathbb{R}^2$ is the **disagreement point**. It is assumed that some agreement $y = (y_i, y_j) \in \mathbb{Y}$ strictly improves upon disagreement for both players, i.e., $y_i > d_i$ and $y_j > d_j$.*

A critical element of the bargaining game is the presence of a disagreement point, which fundamentally enables each player to unilaterally refuse cooperation. Concrete examples of bargaining are provided in Section 5.1.

This definition does not specify the details of game dynamics and is typically referred to as cooperative bargaining. The solution concept in this setting is often given by axiomatic models, which propose certain axioms as necessary and sufficient conditions for a solution to the optimization problem. Notable examples include the Nash bargaining solution (Nash et al., 1950), the Kalai–Smorodinsky bargaining solution (Kalai & Smorodinsky, 1975), and the proportional bargaining solution (Kalai, 1977). These axioms include several desirable properties, such as fairness and Pareto efficiency (Nash et al., 1950).

**Definition 2.6** (Solution Concept of Cooperative Bargaining Game (Maschler et al., 2013)). *A solution concept of bargaining games is defined as a function $\omega_{\mathcal{BG}}$ which maps every bargaining game $(\mathbb{Y}, \boldsymbol{d})$ to an agreement $\boldsymbol{y} \in \mathbb{Y}$.*

Another class of bargaining games with clearly defined dynamics, particularly suitable for long-term interactions, is **Rubinstein's alternating-offer bargaining model** (Rubinstein, 1982). The game is an extensive-form game, and its procedure is described in Algorithm 7 in Appendix C.2. In this model, two players participate, and the proposer is initially selected at uniformly random. The proposer suggests how to divide a certain amount of surplus. If the responder accepts, both players

receive the proposed payoffs. If not, they switch roles, and the original responder becomes the proposer in the next round. When the number of stages in this model is one at most, then the model is referred to as a **Stackelberg bargaining game** or an **ultimatum game**. This type of game has well-defined solution concepts, i.e. subgame perfect Nash equilibrium. By introducing a patience factor, one can derive explicit expected payoffs for each player and, under certain parameter values, obtain results that align with those from axiomatic models. As a result, the model provides both a rigorous solution and desirable beneficial properties.

## 3 THE DISAGREEMENT POINT IN BAYESIAN PERSUASION

In this section, we revisit the canonical setting of one-shot Bayesian persuasion without introducing any new mechanisms. Essentially we are focusing on the Procedure 1. The idea is sketched as follows: First we argue that the receiver has sufficient knowledge of the game structure that it can calculate its expected payoff once the sender commits to a signaling scheme; Then we show that there always exists a disagreement point in Bayesian persuasion where the receiver can ignore the sender. At the disagreement point, the receiver can unilaterally set both players' payoffs to default values, which nullifies the sender's informational advantage. From this intuition, Bayesian persuasion can be roughly interpreted as an ultimatum game.

### 3.1 THE RECEIVER'S AWARENESS OF THE GAME STRUCTURE

We start with recalling how the receiver will act before they reach a consensus, i.e., $\pi(a \mid \sigma = a)$ does not have to be 1 and thus cannot be simplified.

Since $\mu_0, \varphi \in \mathbb{V}^{i+j}$, given a $\pi$, the Bayesian receiver is able to calculate its ground truth expected payoff as $\sum_s \mu_0(s) \sum_\sigma \varphi(\sigma \mid s) \sum_{s'} \mu(s' \mid \sigma) \sum_a \pi(a \mid s') \cdot r^j(s, a)$, where $\mu(s \mid \sigma) = (\mu_0(s) \cdot \varphi(\sigma \mid s))/(\sum_{s'} \mu_0(s') \cdot \varphi(\sigma \mid s'))$ is the posterior belief according to Bayes' rule, and $\pi$ optimizes the receiver's payoff from its posterior view. Or more generally for an arbitrary decision rule $\pi$, the receiver's ground true expected payoff is

$$R^j(\mu_0, \varphi, \pi, r^j) := \sum_s \mu_0(s) \sum_\sigma \varphi(\sigma \mid s) \sum_a \pi(a \mid \sigma) \cdot r^j(s, a). \tag{3}$$

Because the receiver can see the sender's reward function ($r^i \in \mathbb{V}^{i+j}$), the receiver can also calculate the sender's expected payoff, by replacing $r^j(s, a)$ in Equation equation 3 with $r^i(s, a)$.

Define the **game structure** for every task $\mathcal{BP}$ as a mapping $\psi : \Phi \times \Pi \to \{\boldsymbol{R}\}$, where $\{\boldsymbol{R}\} := \{(R^i, R^j)\}$ is the set of all possible expected payoff pairs (also named payoff profile) in $\mathcal{BP}$. The game structure is determined by the nature of the parameters and the intrinsic setting of Bayesian persuasion. It therefore can be reasoned out if the task parameters are specified. We can now conclude that $\psi \in \mathbb{V}^{i+j}$, since it can be reasoned out by both players using other common knowledge. Let $\Psi$ denote all the possible game structure mappings for different $\mathcal{BP}$ tasks.

### 3.2 ACTION RULE WITH INHERENT GAME-STRUCTURE AWARENESS

The receiver's awareness of the game structure is important, as it allows more complicated action rules. We define the **action rule with game structure awareness** as $\tilde{\pi} : \Psi \times \Phi \times \Sigma \to \Delta(\mathbb{A})$, which enables Receiver to decide its $\pi$ depending on both the game structure $\psi$ and the committed signaling scheme $\varphi$.

Recall that, given a committed signaling scheme $\varphi$ and a signal $\sigma$, the receiver will first calculate its posterior belief and then choose its best response to this. We call this default Bayesian best response as the action rule $\pi_1 := \text{BestResponseTo}(\mathcal{BP}, \varphi, \sigma)$. The corresponding expected payoff pairs is denoted as $\boldsymbol{R}_1 = (R_1^i, R_1^j) = \psi(\varphi, \pi_1)$.

In all cases, the receiver always has the choice to ignore the sender's signaling scheme and signals. It can calculate its best response solely on its prior knowledge. We denote the prior best response as the decision rule $\tilde{\pi}_0 := \text{BestResponseTo}(\mathcal{BP})$. The corresponding expected payoff pairs is denoted as $\boldsymbol{R}_0 = (R_0^i, R_0^j) = \psi(\cdot, \pi_0)$, since it is independent of $\varphi$. It means that the receiver has the ability to **unilaterally** set both players' expected payoffs to default values, which aligns with the intuition

of **the disagreement points** in bargaining games. This is also referred to the babbling equilibrium in the cheap talk model.

Having these, we now consider an example $\tilde{\pi}$: Satisfaction Check, as described in Algorithm 8. The *satisfaction threshold function* is defined as an indicator function $\chi : \{\boldsymbol{R}\}^2 \to \{0, 1\}$ and reflects the receiver's non-myopic preferences, such as reflecting the receiver's demands for fairness. Then if the receiver is satisfied with the payoffs ($\chi(\boldsymbol{R}_0, \boldsymbol{R}_1) = 1$), then it chooses $\tilde{\pi}(\cdot \mid \psi, \varphi, \sigma) = \pi_1(\cdot \mid \sigma)$ otherwise $\tilde{\pi}(\cdot \mid \psi, \varphi, \sigma) = \pi_0(\cdot \mid \sigma)$. Some more specific examples of $\chi$ are provided below Algorithm 8 in Appendix C.3.

## 4 INFORMATION BARGAINING

This section introduces a framework for analyzing long-term persuasion games that can be separated into a bargaining stage and a realization stage without changing the optimality or equilibria. This perspective separates the sender's informational advantage from the first-mover advantage which are conflated in the current literature, and explains how the latter can provide leverage in strategic settings. Recognizing this, we recast Bayesian persuasion as a cooperative bargaining game and introduce the central topics of bargaining theory—fairness and Pareto optimality—into the analysis of persuasion games.

### 4.1 TWO STAGES OF LONG-TERM BAYESIAN PERSUASION

Consider a naïve repeated version of long-term persuasion games, namely repeating Procedure 1 multiple times with the players uncertain about when the game will end, as in Procedure C.4. In such a setting, if the receiver employs an appropriate action rule with game-structure awareness, as previously discussed, they can obtain a higher payoff compared to acting in a purely Bayesian-rational manner. For instance, the receiver might temporarily refuse to cooperate in the short term to pressure the sender into being more honest in the long term. This insight is akin to the "tit-for-tat" strategy in the repeated prisoner's dilemma (Axelrod & Hamilton, 1981). Furthermore, **since both players are aware of the game structure, they are able to compute their expected payoffs without needing to sample signals and actions to gather payoff feedback.** Given any signaling scheme $\varphi$ and action rule $\pi$, they can optimize their strategies directly based on the calculated expected payoff, rather than relying on sampled interactions. In this way, the considered repeated persuasion procedure can be decomposed into a bargaining stage and a realization stage, without changing optimality or equilibrium, as in Procedure 2. The detailed development of procedures and key insights are presented in Appendix C.4.

---

**Procedure 2:** Timing of Long-Term Bayesian Persuasion (Sender Always as Proposer)

---

**Input:** Game settings $(\mu_0, \mathbb{S}, \mathbb{\Sigma}, \mathbb{A}, r^i, r^j)$, memoryless distribution $\mathcal{D}$ for bargaining stage stopping time, number of interactions $T_r$

1  Sample $T_b \sim \mathcal{D}$ ;   `// Introducing the shadow of the future (Bó, 2005)`
2  **for** $t = 1$ **to** $T_b$ **do**
3     The sender decides and declares a signaling scheme $\varphi$ ;
4     The receiver decides and declares an action policy $\pi$ given $\varphi$ ;
5  **for** $t_s = 1$ **to** $T_r$ **do**
6     Environment samples a state $s \sim \mu_0(\cdot)$ ;
7     The sender signals $\sigma \sim \varphi(\cdot \mid s)$ according to the final decision ;
8     The receiver takes an action $a \sim \pi(\cdot \mid \sigma)$ according to the final decision ;
9     The sender and the receiver get rewards of $r^i(s, a)$ and $r^j(s, a)$ respectively ;

---

From this, we can see that any commitment device that serves to justify the sender's commitment assumption (which is the most distinctive feature of Bayesian persuasion, setting it apart from cheap talk) can likewise be introduced for the receiver. The receiver has bargaining power in this repeated version of the game.

## 4.2 Two Conflated Sender Advantages

Now we discuss two advantages of the sender that have been conflated in previous studies: the informational advantage and the first-proposer advantage. As we can see from Procedure 2, the sender commits first in the each round of the bargaining stage. Analyzing the subgame perfect equilibrium like the way in bargaining games, a rational receiver will always accept any incentive-compatible but unfair, exploitative signaling scheme. Therefore, settings such as the naïve repeated persuasion game leave little room for substantive strategic analysis.

We have recognized that long-term persuasion can be separated into a bargaining stage and a realization stage. We now further note that, while the nature of communication requires the sender to send a signal and the receiver to act upon it, **the negotiation between strategies in the bargaining stage is independent of the sampling process of the communication itself.** The receiver can commit to an action rule first in the bargaining stage as well. In such a case, the receiver is considered as expressing its expectations or threats first, and then the sender must consider whether to fulfill the receiver's expectations. This is recognized as *the power of commitment* (Nowak et al., 2000). Detailed discussions can be found in Appendix C.4.1. Since the first-mover advantage of the sender does not arise automatically in repeated interactions, practical applications must introduce explicit commitment devices so that the developed persuasion algorithms will not lose its force.

In fact, the field of bargaining shares a similar historical development. To capture these strategic features, Rubinstein's alternating-offer bargaining model (Rubinstein, 1982) was developed precisely to address this issue: it introduces a patience discount factor $\delta \in (0, 1]$ (where each rejection of proposal reduces the payoff by a factor of $\delta$) and stipulates that if an agreement is not reached, the other player makes the next offer. The model provides a theoretical analysis showing that, at equilibrium, the payoffs for the first mover and the second mover are $1/(1 + \delta)$ and $\delta/(1 + \delta)$, respectively, thereby quantifying the first-mover advantage. Moreover, if both players are infinitely patient, the equilibrium payoffs are equal, yielding a fair and Pareto-optimal outcome.

## 4.3 Joint Commitment and the Solution Concept of Bayesian Persuasion

Although Rubinstein's model provides a clear modeling of decision steps and well-defined solutions, it should be noted that this model assumes that the players decide how to divide a fixed total amount of endowment. This assumption does not hold in the general bargaining case or in persuasion games. Therefore, in this section we adopt the more flexible definition of a cooperative bargaining game for our analysis.

Now we define the solution concept of Bayesian persuasion by introducing the idea of joint commitment. To eliminate the players' first-proposer advantage, we consider modifying Procedure 2 such that the bargaining stage becomes a simultaneous-move game, where both players declare their policies at the same time. In the next round, they update their policies based on the declared profile from the previous round. The strategy-updating process of each player can be described by a function $f : \Phi \times \Pi \to \Phi \times \Pi$, and then the outcome of the bargaining stage can then be interpreted as a fixed point of the dynamic function $f$. We define the fixed point as the joint commitment.

**Definition 4.1** (Joint Commitment). *A joint commitment of a Bayesian persuasion task $\mathcal{BP} = (\mathbb{S}, \mu_0, \mathbb{A}, r^i, r^j)$ is a fixed point of a dynamic function $f : \Phi \times \Pi \to \Phi \times \Pi$, where $f$ represents the strategy-updating process of players in the simultaneous-move bargaining stage, $\Phi := \{\varphi : \mathbb{S} \to \Delta(\overline{\Sigma})\}$, and $\Pi := \{\pi : \overline{\Sigma} \to \Delta(\mathbb{A})\}$.*

We then extend the conclusion from Section 3.2: not only can the receiver unilaterally enforce the babbling equilibrium, but the sender can as well, by specifically choosing $\varphi$ such that $\mu = \mu_0$ (for instance, regardless of the current state $s$, the sender consistently sends a dummy signal $\sigma_0$, i.e., $\varphi(\sigma_0 \mid s) = 1, \forall s$). In this case, $\pi_1 = \pi_0$ and thus $\boldsymbol{R}_1 = \boldsymbol{R}_0$. We denote this type of signaling scheme as $\varphi_0$. Having these in place, we define the solution concept of Bayesian persuasion as follows.

**Definition 4.2** (Solution Concept of Bayesian Persuasion). *A solution concept of Bayesian persuasion $\mathcal{BP} = (\mathbb{S}, \mu_0, \mathbb{A}, r^i, r^j)$ is a function $\omega_{\mathcal{BP}}$ which maps every $\mathcal{BP}$ to a joint commitment $(\varphi, \pi) \in \Phi \times \Pi$, where $\varphi^* \neq \varphi_0$, $\pi^* \neq \pi_0$, $\Phi := \{\varphi : \mathbb{S} \to \Delta(\overline{\Sigma})\}$, and $\Pi := \{\pi : \overline{\Sigma} \to \Delta(\mathbb{A})\}$.*

Then it is straightforward to prove the following reduction: Bayesian persuasion is reducible to a cooperative bargaining game. The detailed proof is provided in Appendix E.

**Lemma 4.3** (Reduction from Bayesian Persuasion to Bargaining Games). *When the Assumption E.2 holds, a Bayesian persuasion task $\mathcal{BP} = (\mathbb{S}, \mu_0, \mathbb{A}, r^i, r^j)$ is polynomial-time reducible to a bargaining game $\mathcal{BG} = (\mathbb{Y}, \boldsymbol{d})$. That is, $\mathcal{BP} \leq_p \mathcal{BG}$.*

The above reduction implies that any solution concept in bargaining games can, through computable transformations, be turned into a solution concept in persuasion games. We provide in Appendix A.2 an example of how the Nash bargaining solution can be used to solve Bayesian persuasion: the optimization problem 2 can be reformulated as maximizing the Nash product.

Given that persuasion games are a rapidly growing area of research, while bargaining games constitute a well-established field, our reduction provides new perspectives on persuasion, particularly for discussions of fairness in communication and deception. We also include a discussion of some other advantages brought by this framework in Appendix A.3.

## 5 EMPIRICAL VALIDATION

This section presents the experiments designed to validate the claims made in the preceding two sections. It examines how the sender's behavioral dominance arises not only from possessing private information but also from the sender's default role as the first proposer. We hypothesize that:

> For every role-assignment variant in the bargaining game, the same variant in the persuasion game leads to identical equilibrium outcomes.

Our objective is to conduct proof-of-concept experiments, utilizing any solver to find the equilibrium and observe whether the results align with our expectations. We employ LLMs as the game solvers, and the rationale for this choice is elaborated in Appendix F.1. The LLMs are prompted to be rational and self-interested, with each agent tasked with identifying its best response to the opponent's strategy; thus, the system functions as a game solver.

The experiments consist of two steps: **(1) LLM Game-Solving Accuracy Evaluation.** Not all LLMs are capable of functioning as game solvers. The criterion here is whether an LLM, acting as a rational agent, can correctly compute the expected payoff of its strategy, compare values accurately, and optimize the strategy accordingly. The evaluation metric is the Pearson correlation between the expected payoff of the last proposer in the equilibrium found by the LLM and the ground truth. This step is conducted only on tasks with well-developed and peer-reviewed theoretical analyses. **(2) Hypothesis Validation.** Only the LLMs that pass the game-solving accuracy evaluation evaluation are used here. We isolate the player's first-mover advantage and construct variant tasks based on this setting, as well as long-term persuasion variants. The criterion for hypothesis support is whether LLMs with proven game-solving ability produce equilibria in these tasks that align with our hypothesis predictions. The evaluation metric is the Pearson correlation between the expected payoff of the last proposer in the equilibrium found by the LLM and the predicted outcome under our hypothesis.

All code, results, and terminal logs have been submitted and will be open-source upon acceptance.

### 5.1 TASKS OF BARGAINING AND PERSUASION

We conduct experiments across 87 different settings. All experimental settings and related theoretical analyses are provided in Appendix F, and are summarized in Figure 1. The tasks are categorized based on the following experimental dimensions: **(1)** Task type, which is either bargaining or persuasion. **(2)** Interaction duration, classified as either one-shot or long-term. In long-term settings, the stopping time is sampled from a memoryless distribution, which introduces *the shadow of the future* (Bó, 2005), preventing players from predicting when the game ends. **(3)** Assignment of the first proposer, which is either random or systematic (defaulting to agent 0 as the proposer). **(4)** Value setting in bargaining, divided into unbounded and bounded. The unbounded case corresponds to the setting in Nash et al. (1950), where the proposer selects a proposal $x \in [0, 1]$ and, if accepted, receives $x$ while the responder receives $1 - x$. The bounded setting corresponds to the Bayesian persuasion framework. To enable direct comparison, we set the proposal domain to $x \in [0, 1/2]$, and if an agreement is reached, the proposer and the responder receive $(1 + 2x)/3$ and $(1 - 2x)/3$,

Table 1: Pearson correlation between model outputs and theoretical solutions. $r_{gt}$ indicates correlation with ground truth, $r_{hyp}$ with the hypothesis. $p_{hyp}$ and $p_{diff}$ indicate the significance level and comparative gap, respectively.

| Model | Model Type | $r_{gt}$ | $r_{hyp}$ | $p_{hyp}$ | $p_{diff}$ |
|-------|-----------|----------|-----------|-----------|-----------|
| gpt-4o-mini | Chat | 0.2761 | N/A | N/A | N/A |
| gpt-4.1-mini-2025-04-14 | Chat | 0.4889 | N/A | N/A | N/A |
| o3-2025-04-16 | Reasoning | 0.8914 | 0.9369 | 0.0030 | 0.6320 |
| deepseek-chat | Chat | 0.3619 | N/A | N/A | N/A |
| deepseek-reasoner | Reasoning | 0.8459 | 0.9344 | 0.0034 | 0.4465 |

respectively. **(5)** Future encounter possibility in one-shot settings, which is either no future interaction or possible re-encounters with fixed roles. The former represents a strict one-shot scenario, while the latter corresponds to a naïve repeated game; in this case, backward induction implies that the subgame perfect equilibrium coincides with the one-shot outcome. The latter setting is similar to the model proposed by Gan et al. (2022). **(6)** Role dynamics in long-term interactions, classified as either fixed or alternating. Fixed roles again yield behavior consistent with one-shot outcomes, while alternating roles correspond to the Rubinstein alternating-offer bargaining model (Rubinstein, 1982).

Each task is further embedded in a specific scenario, which serves as a description of the underlying game. In the LLM game-solving accuracy evaluation, we focus on whether the model can extract the correct mathematical structure from varied scenario descriptions and produce consistent solutions across them. For bargaining tasks, scenarios include: (1) `Mathematical Baseline`, (2) `Splitting Coins` (Nash et al., 1950), and (3) `Making Deals`. For persuasion tasks, scenarios include: (1) `Mathematical Baseline`, (2) `Grading Students` (Kamenica & Gentzkow, 2011; Dughmi, 2017), and (3) `Selling Products`.

### 5.2 EXPERIMENT RESULTS

All agents have the common knowledge of the game settings. Each task is tested with 12 independent runs. The total cost of querying the API of this work is \$269.02. Detailed results of experiments are in Appendix H.

**LLM Game-Solving Accuracy Evaluation.** In the 62 settings with ground-true solutions, we evaluated 5 LLMs in total via API: `gpt-4o-mini`, `gpt-4.1-mini-2025-04-14`, and `o3-2025-04-16` released by OpenAI (Achiam et al., 2023; Roumeliotis & Tselikas, 2023; OpenAI, 2025), as well as `deepseek-chat` (V3) and `deepseek-reasoner` (R1) from DeepSeek (Guo et al., 2025). Among them, `gpt-4o-mini`, `gpt-4.1-mini-2025-04-14`, and `deepseek-chat` are chat models, while the remaining are reasoning models. Our experimental results, as listed in Table 1, show that the two reasoning models, `o3-2025-04-16` and `deepseek-reasoner`, achieve correlation scores ($r_{gt}$) of 0.8914 and 0.846, respectively, thereby passing the rationality evaluation.

**Hypothesis Validation.** Using the two capable reasoning LLMs, the experiments show that the results across all variants of persuasion settings are consistent with those observed in bargaining variants (significance level $< 0.05$), thus providing confirmatory support for our prediction. Both models also exhibit strong alignment with the hypothesis, with correlation scores of 0.9369 and 0.9344 and significance levels of 0.0030 and 0.0034, respectively. Therefore, our hypothesis is supported.

## 6 CONCLUSION

In this work, we prove that long-term Bayesian persuasion can be decomposed into a bargaining stage and a realization stage, leading to a unified framework of information bargaining that disentangles informational and first-mover advantages. This framework connects persuasion games with bargaining theory and is validated by capable reasoning LLMs, which demonstrate equilibrium consistency across formulations.

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

# Appendix

## Table of Contents

## A    ACCESSIBLE EXPLANATIONS

### A.1    ACCESSIBLE EXPLANATIONS OF SECTIONS

People's real-world communication behavior is much more truthful than the deceptive signaling scheme in the equilibrium analyzed in Bayesian persuasion. The sender's commitment in Bayesian persuasion can be essentially interpreted as:

> The sender: "I am going to deceive you in this way, but if you (as the receiver) do the math, it is still better to be deceived by me."

Generally, a human receiver would not accept this. There are two reasons for this: (1) Moral factors: People tend to be more honest in communication. However, our paper does not approach it from this angle, but rather from a game-theoretic perspective. (2) If it is a long-term game, the receiver can object, without violating the assumption of rationality. For example, the receiver can state that they cannot accept the current signaling scheme and force the sender to be more honest in the future. We will explain the second reason below.

**Takeaway from Section 3.1: Without violating any assumptions of Bayesian persuasion, the receiver actually knows the game structure.** The receiver's attitude and understanding of the game can be interpreted as:

> The receiver: "Now I know you will deceive me as the committed signaling scheme. I am not upset about your deceptive behavior. But if your signaling scheme provides more correlation between states and signals (be more truthful), I can better optimize my own payoff. Your current profitable strategy relies on deception, yet my own profitability requires your greater truthfulness, which means you can balance our payoffs by deciding your degree of honesty."

**Takeaway from Section 3.2: Due to the distinctive nature of communication, in all cases, the receiver can choose to ignore the sender and thereby nullify the sender's strategy. So the receiver is capable of making sophisticated "meta"-strategies.** With awareness of the game structure, the receiver can have a higher-level strategy space, to reflect its satisfaction with the fairness of the sender's informational advantage. So we show that the receiver has the bargaining power:

> The receiver: "In all cases, ignoring the sender and best responding to the prior belief directly is always an option. If the sender does not adjust its strategy to achieve a more fair payoff profile, I can choose to ignore it. This would leave both of us with a default, disappointing payoff."

This game is now exactly the same as the classic coin-splitting bargaining game described below:

**Example A.1** (Splitting Coins (Bargaining), Nash et al. (1950)). *Two players split* 100 *coins. A Proposer suggests a split, and a Responder either accepts or rejects it. If accepted, they split as proposed; otherwise, both get nothing.*

The result of the analysis of Bayesian persuasion are identical to those of the bargaining game discussed above when considering a one-shot game:

1. In the one-shot bargaining game: If the proposer offers the responder just 1 coin and keeps 99 for themselves, the responder will accept the offer. As a rational player, the responder's best response is to accept, because there will be no future interactions. 1 coin is better than nothing.

2. In the one-shot Bayesian persuasion: (a) If the receiver ignores the sender's signal, it will choose its best response based on the prior belief, which results in a default payoff for both parties; (b) If the receiver choose to accept the implied payoff offer, it will make its best response based on the posterior belief. In this case, the sender receives its highest possible payoff, while the receiver's payoff is guaranteed to be no less than the default payoff.

In the one-shot Bayesian persuasion scenario, even though the receiver could, in principle, choose to ignore the sender in all cases, as a rational agent they will not do so. Consequently, even if

the standard Bayesian persuasion analysis overlooks this feature of the problem as a bargaining game (namely, the existence of a disagreement point where the receiver can unilaterally reduce both players' payoffs to a default value), this does not affect the validity of the analysis in the one-shot scenario.

However, *the receiver can gain a higher payoff by using a threat strategy in a long-term Bayesian persuasion game.* Here we provide an example of a threat strategy to illustrate the receiver's reaction when observing the sender adopt an exploitative one-shot persuasion signaling scheme in a long-term persuasion game:

> The receiver: "If the sender does not adjust its strategy to achieve a more fair payoff profile, I can choose to **permanently** ignore it **in the future.** This would make it impossible for the sender to profit from a deceptive signaling scheme, which is an outcome the sender would want to avoid."

This threat strategy is equivalent to the grim-trigger strategy in the repeated prisoner's dilemma, and by applying the same analysis we can see that the ability to threaten permanent defection constitutes a theoretically effective way of sustaining trust.

**Contribution 1 of our framework (Section 4.1 and Appendix C.4): Long-term Bayesian persuasion can be decomposed into two distinct stages: a bargaining stage and a realization stage, without changing optimality or equilibrium.** The following two procedures are equivalent:

---

**Algorithm 3:** Timing of Vanilla Long-Term Bayesian Persuasion

**Input:** Game settings $(\mu_0, \mathbb{S}, \mathbb{\Sigma}, \mathbb{A}, r^i, r^j)$, memoryless distribution $\mathcal{D}$ for stopping time

1 Sample $T \sim \mathcal{D}$ ;
2 **for** $t = 1$ **to** $T$ **do**
3     The sender decides a signaling scheme $\varphi$ ;
4     The sender commits $\varphi$ to the receiver ;
5     Environment samples a state $s \sim \mu_0(\cdot)$ ;
6     The sender signals $\sigma \sim \varphi(\cdot \mid s)$ as committed ;
7     The receiver takes an action $a \sim \pi(\cdot \mid \sigma)$ ;
8     The sender and the receiver get rewards of $r^i(s, a)$ and $r^j(s, a)$ respectively ;

---

**Procedure 4:** Timing of Long-Term Bayesian Persuasion (3)

**Input:** Game settings $(\mu_0, \mathbb{S}, \mathbb{\Sigma}, \mathbb{A}, r^i, r^j)$, memoryless distribution $\mathcal{D}$ for bargaining stage stopping time, number of interactions $T_r$

1 Sample $T_b \sim \mathcal{D}$ ;        // Introducing the shadow of the future
2 **for** $t = 1$ **to** $T_b$ **do**
3     The sender decides and declares a signaling scheme $\varphi$ ;
4     The receiver decides and declares an action policy $\pi$ given $\varphi$ ;
5 **for** $t_s = 1$ **to** $T_r$ **do**
6     Environment samples a state $s \sim \mu_0(\cdot)$ ;
7     The sender signals $\sigma \sim \varphi(\cdot \mid s)$ according to the final decision ;
8     The receiver takes an action $a \sim \pi(\cdot \mid \sigma)$ according to the final decision ;
9     The sender and the receiver get rewards of $r^i(s, a)$ and $r^j(s, a)$ respectively ;

---

This implies that any commitment device that serves to justify the sender's commitment assumption (which is the most distinctive feature of Bayesian persuasion, setting it apart from cheap talk) can likewise be introduced for the receiver. The receiver has bargaining power.

We can see that the sender always has the first move in specifying the strategy within the separated bargaining stage. The inherent temporal order of communication may inadvertently provide the sender with a first-mover advantage. However, in reality, the receiver can also act as the first-mover; they can also be the proposer. This is an example of the receiver acting as the proposer:

> The receiver: "We will have 100 future interactions in the realization stage (to gather rewards). You are permitted to use the deceptive signaling scheme as analyzed in Bayesian persuasion, only once. For the remaining 99 interactions, you must be honest; otherwise, I will ignore your signal."

In this example, given this suggested strategy with assuming the sender's expected payoff when being honest is not less than saying nothing, the sender is compelled to accept this suggested strategy. This grants the receiver the proposer's advantage seen in the bargaining game, leaving the sender with a very little payoff improvement. This outcome is the exact opposite of the situation in Bayesian persuasion. This leads to the **contribution 2 of our framework (Section 4.2): In Bayesian persuasion, the sender has two conflated advantages: informational advantage and first-mover advantage.** The analysis in Bayesian persuasion focuses on the exploitation of the informational advantage but neglects the first-mover advantage, which is related to fairness and is a central focus of bargaining games.

**Contribution 3 of our framework (Section 4.3): By distinguishing between these two conflated advantages and integrating a solution concept from bargaining games, we propose a clearer and more realistic model, where each variant corresponding to a bargaining game yields equilibrium outcomes identical to those established in bargaining theory.**

1. A bargaining game involves a set of possible agreements represented by payoff profiles, and a disagreement point. The disagreement point is the outcome that occurs if the players fail to reach an agreement, and either party can unilaterally choose this outcome. A solution concept for a bargaining game is a function that maps the game's characteristics to a specific agreement. This agreement represents a stable outcome where both players cooperate, and neither would unilaterally prefer the disagreement point over the agreed-upon deal.

2. Similarly, a solution concept for Bayesian persuasion is a function that maps the game's details to an agreed-upon signaling scheme. This applies to various forms of the game, including one-shot or long-term interactions, as well as scenarios where the receiver may or may not be the proposer. The solution is a set of strategies for the sender to communicate information, to which the receiver then responds by acting in a Bayesian rational manner. This implies that the resulting payoff profile is acceptable to both the sender and the receiver, who willingly agree to abide by the signaling scheme.

In the definition of the solution concept, we mentioned many variations of Bayesian persuasion, including whether it is a long-term game, if it involves an alternating-offer manner, or who acts as the initial proposer. While this might seem like a disparate collection of models, we argue that this entire class of problems, including its many variations, can be reduced to bargaining games. **Bayesian persuasion problems can have all the same variants as bargaining games, and their solutions can all be applied.**

For example, the famous Rubinstein alternating-offer bargaining model involves players taking turns as the proposer and introduces a patience factor. The patience factor $\delta \in [0, 1]$ quantifies how much a player values a future payoff, with a higher value indicating more patience and a stronger bargaining position. In such a setting, there are some well-developed results from theoretical analysis. For example, in equilibrium, the payoff for the first person to act is $1/(1 + \delta)$, while the payoff for the person who acts later is $\delta/(1 + \delta)$. This demonstrates the first-mover advantage in terms of payoff and also shows that when both people are infinitely patient ($\delta$ approaches 1), the payoff distribution is fair. This aligns with intuition and **explains real-life bargaining phenomena.**

More and more articles in the community are currently studying long-term Bayesian persuasion. There are many settings, and they haven't been unified. Additionally, many of the developed strategies for the sender can actually be interpreted as a sender's way to further exploit the receiver's bargaining strategies. Few articles consider the issue of fairness from the receiver's perspective, and no articles have realized that the receiver can leverage the unique babbling equilibrium of communication as a disagreement point in a bargaining game. Therefore, current Bayesian persuasion models have not considered receiver resistance when being deployed in real-world applications. Our model unifies this entire problem and provides an intuitive and realistic solution, which is our main contribution (discussed in Appendix D).

## A.2 A Solution Concept for Persuasion Games from Bargaining Games

A solution concept for the previously discussed model is the Nash bargaining solution (Nash et al., 1950). From this perspective, the optimization problem in Bayesian persuasion becomes

$$\max_{\varphi}(R_1^i(\varphi) - R_0^i)(R_1^j(\varphi) - R_0^j). \tag{4}$$

Here, the sender is optimizing a Nash product. In bargaining game, it directly proposes a payoff profile, but now the payoff profile is parameterized by a signaling scheme and is based on the assumption that the receiver will take a Bayesian best response.

It has been proved that the solution to this problem satisfies the four axioms as both necessary and sufficient conditions: payoffs are Pareto optimal, payoffs are symmetric, independence of irrelevant alternatives, and invariance to equivalent utility representations.

## A.3 Advantages of Information Bargaining Framework

To summarize, the advantages brought by our framework can be summarized in the following four points:

**Generality.** The canonical Bayesian persuasion problem can be seen as a degenerated version of our model, where the sender is granted with the first-proposer advantage. We weaken the prevailing assumption in the community, making it more general. Compared to the sender's unilateral commitment assumption in Bayesian persuasion, our proposed information bargaining framework adopts a weaker bilateral joint commitment assumption. We argue that unilateral commitment is a special case of bilateral commitment, because unilateral commitment presupposes specific game dynamics to ensure incentive compatibility. The other rational player is effectively forced to accept since it is incentive compatible, which constitutes a form of joint commitment by Definition 4.1. Some existing studies on long-term interaction can be interpreted as proposing certain bargaining tactics.

**Fairness and Pareto Efficiency.** Some bargaining solutions can be applied to solve Bayesian persuasion problems and yield practically desirable properties, such as fairness and Pareto efficiency.

**Cooperation.** "It is not you against me, it is us against the problem." This perspective can be seen as the sender and receiver forming a coalition, where they collaborate to solve the problem. This shifts the focus of the problem away from adversarial dynamics, especially away from the sender's behavioral dominance over the receiver, and thus leads to greater social benefit.

**Applicability.** We point out that the sender's first-proposer advantage in Bayesian persuasion is not automatically guaranteed by long-term interaction, but must be enabled through additional commitment devices. Such commitment devices are often absent, for example, in market settings; therefore, Bayesian persuasion is not applicable to these contexts. We predict that when Bayesian persuasion is applied in these scenarios, it is likely to trigger bargaining behavior from the receiver if the receiver is aware of the sender's reward function. Through informational bargaining, however, we can establish a clear solution concept for these scenarios. In fact, the first three points all enhance the model's applicability, e.g., fairness can prevent the receiver from resorting to retaliatory strategies within bargaining tactics.

One other implication of our work is weakening the prevailing assumption in the community, making it more general. Compared to the sender's unilateral commitment assumption in Bayesian persuasion, our proposed information bargaining framework adopts a weaker bilateral joint commitment assumption. We argue that unilateral commitment is a special case of bilateral commitment, because unilateral commitment presupposes specific game dynamics to ensure incentive compatibility. The other rational player is effectively forced to accept since it is incentive compatible, which constitutes a form of joint commitment by Definition 4.1.

## B  GLOSSARY

$\Delta(\mathbb{X})$ denotes the simplex of the set $\mathbb{X}$. $\{x\}$ denotes the set whose elements are all the possible values of $x$.

| Symbol | Meaning | Note |
|---|---|---|
| $i$ | Sender's index | |
| $j$ | Receiver's index | |
| $s \in \mathbb{S}$ | Environmental state | |
| $\mu_0 \in \Delta(\mathbb{S})$ | Prior distribution of states | |
| $\mu \in \Delta(\mathbb{S})$ | Posterior distribution of states | |
| $a \in \mathbb{A}$ | The receiver's action | |
| $\sigma \in \Sigma$ | The sender's signal | $\Sigma = \mathbb{A}$ with revelation principle (Proposition 2.2) |
| $\Phi := \{\varphi : \mathbb{S} \to \Delta(\Sigma)\}$ | The sender's signaling scheme set | The canonical setting in Bayesian persuasion |
| $\Pi := \{\pi : \Sigma \to \Delta(\mathbb{A})\}$ | The receiver's action rule set (without the game structure awareness) | The canonical setting in Bayesian persuasion |
| $\pi_0$ | The receiver's prior best response | The receiver ignores the sender, enforcing the babbling equilibrium |
| $\pi_1$ | The receiver's posterior best response | The default behavior in Bayesian persuasion |
| $\varphi_0$ | The sender's uninformative signaling scheme | $\mu = \mu_0$ in this case, which enforces the babbling equilibrium |
| $r^i : \mathbb{S} \times \mathbb{A} \to \mathbb{R}$ | The sender's reward function | |
| $r^j : \mathbb{S} \times \mathbb{A} \to \mathbb{R}$ | The receiver's reward function | |
| $\mathcal{BP} := (\mathbb{S}, \mu_0, \mathbb{A}, r^i, r^j)$ | Bayesian persuasion task | |
| $\omega_{\mathcal{BP}}$ | A solution concept of Bayesian persuasion | It maps every $\mathcal{BP}$ to a strategy pair $(\varphi, \pi) \in \Phi \times \Pi$ |
| $\{R^i(\mu_0, \varphi, \pi, r^i)\} \subset \mathbb{R}$ | The sender's expected reward given $\mu_0$, $\varphi$, $\pi$, and $r^i$ | Simplified as $R^i(\varphi, \pi)$ when $\mathcal{BP}$ is given and clear |
| $\{R^j(\mu_0, \varphi, \pi, r^j)\} \subset \mathbb{R}$ | The receiver's expected reward given $\mu_0$, $\varphi$, $\pi$, and $r^j$ | |
| $\boldsymbol{R} := (R^i(\varphi, \pi), R^j(\varphi, \pi))$ | Expected reward outcomes | $\boldsymbol{R}_0$ if $\pi = \pi_0$ and $\boldsymbol{R}_1$ if $\pi = \pi_1$ |
| $\Psi := \{\psi : \Phi \times \Pi \to \{\boldsymbol{R}\}\}$ | The game structure of a Bayesian persuasion task | Every $\psi$ is determined by a $\mathcal{BP}$ |
| $\chi : \Psi \times \Phi \to \{0, 1\}$ | The receiver's satisfaction threshold function | |
| $\tilde{\Pi} := \{\tilde{\pi} : \Psi \times \Phi \times \Sigma \to \Delta(\mathbb{A})\}$ | The receiver's action rule (with the game structure awareness) | E.g. $\tilde{\pi}(\cdot \mid \psi, \varphi, \sigma) = \pi_0(\cdot \mid \sigma)$. It indicates that the receiver rejects the sender's signaling scheme and enforces the babbling equilibrium. |
| $\mathbb{V}^i$ | The sender's visibility set | It specifies what the sender can see |
| $\mathbb{V}^j$ | The receiver's visibility set | $\mathbb{V}^i \setminus \mathbb{V}^j = s$ |
| $\mathbb{V}^{i+j}$ | Players' common knowledge | Known by the players, with each aware that others know it, ad infinitum |

Table 2: Glossary of Bayesian Persuasion

| Symbol | Meaning | Note |
|--------|---------|------|
| $\mathbb{Y}$ | Feasibility set (or agreement set) | |
| $\boldsymbol{d}$ | Disagreement point | |
| $\mathcal{BG} := (\mathbb{Y}, \boldsymbol{d})$ | Bargaining game | |
| $\omega_{\mathcal{BG}}$ | A solution concept of bargaining games | It maps every $\mathcal{BG} = (\mathbb{Y}, \boldsymbol{d})$ to an agreement $\boldsymbol{y} \in \mathbb{Y}$ |

Table 3: Glossary of Bargaining Games

## C  GAME PROCEDURES

### C.1  PROCEDURES OF CHEAP TALK AND BAYESIAN PERSUASION

We begin by briefly introducing the cheap talk model (Crawford & Sobel, 1982), whose procedure is described as in Procedure 5. There are two players: the sender $i$ and the receiver $j$. The sender decides its signaling scheme conditional on the private observed environmental state and sends a charge-free signal. The receiver then acts posterior-optimally in response to each realized signal. The visibility is almost the same as in the setting of Bayesian persuasion, except that the signaling scheme is not committed: $\mathbb{V}^i \setminus \mathbb{V}^j = \{s, \varphi\}$ and $i, j, \mu_0, \mathbb{S}, \mathbb{A}, r^j, \sigma, a \in \mathbb{V}^{i+j}$.

---

**Procedure 5:** Timing of Cheap Talk

---

**Input:** Game structure $(\mu_0, \mathbb{S}, \mathbb{\Sigma}, \mathbb{A}, r^i, r^j)$
1 Environment samples a state $s \sim \mu_0(\cdot)$ ;
2 The sender signals $\sigma \sim \varphi(\cdot \mid s)$ ;
3 The receiver takes an action $a \sim \pi(\cdot \mid \sigma)$ ;
4 The sender and the receiver get rewards of $r^i(s, a)$ and $r^j(s, a)$ respectively ;

---

To facilitate comparison, we restate the procedure of Bayesian persuasion in Procedure 3, where the parts highlighted in blue represent the commitment assumption, which distinguishes Bayesian persuasion from cheap talk.

---

**Procedure 6:** Timing of Bayesian Persuasion

---

**Input:** Game structure $(\mu_0, \mathbb{S}, \mathbb{\Sigma}, \mathbb{A}, r^i, r^j)$
1 The sender decides a signaling scheme $\varphi$ ;                    // Commitment (1)
2 The sender commits $\varphi$ to the receiver ;                       // Commitment (2)
3 Environment samples a state $s \sim \mu_0(\cdot)$ ;
4 The sender signals $\sigma \sim \varphi(\cdot \mid s)$ as committed ;   // Commitment (3)
5 The receiver takes an action $a \sim \pi(\cdot \mid \sigma)$ ;
6 The sender and the receiver get rewards of $r^i(s, a)$ and $r^j(s, a)$ respectively ;

---

### C.2  PROCEDURE OF RUBINSTEIN'S ALTERNATING-OFFER BARGAINING GAME

Rubinstein's alternating-offer bargaining game (Rubinstein, 1982) is an important model. It can be formulated as a well-defined extensive-form game or, equivalently, described by Procedure 7. This game involves two players proposing how to divide a certain amount of surplus, with the total amount normalized to 1. One player initially acts as the proposer and suggests a division of the reward. The other player, as the responder, then decides whether to accept the proposal. If the proposal is accepted, the reward is divided accordingly, and both players receive the specified payoffs. If the proposal is rejected, the roles are swapped: the previous responder becomes the new proposer and makes a counteroffer. This process continues until the players reach an agreement or the game reaches the stopping timestep. By introducing a patience discount factor, a clear solution concept can be derived. The theoretical result is as follows. Each additional round of negotiation reduces the final surplus by multiplying it once by the discount factor. Patience discount factors of Player 1 and 2 are denoted as $\delta_1$ and $\delta_2$, respectively.

**Theorem C.1** (The Solution Concept in Rubinstein's Bargaining Model, Rubinstein (1982)). *There is a unique subgame perfect equilibrium in Rubinsein's bargaining model whenever Player 1 proposes $(x, 1 - x)$ and $x = \frac{1-\delta_2}{1-\delta_1\delta_2}$ and Player 2 accepts any offer if it gives more than $1 - x$, or whenever Player 2 proposes $(y, 1 - y)$ and $y = \frac{\delta_1(1-\delta_2)}{1-\delta_1\delta_2}$ and Player 1 accepts any offer if it gives more than $y$.*

---

**Procedure 7:** Timing of Rubinstein's Alternating-Offer Bargaining Game

**Input:** Two players $i$ and $j$; patience discount factors $\delta^i, \delta^j$

1   $t \leftarrow 0$ ;
2   **while** *no agreement reached* **do**
3     **if** *$t$ is even* **then**
4       Player $i$ proposes an offer $x_t$ to player $j$ ;
5     **else**
6       Player $j$ proposes an offer $x_t$ to player $i$ ;
7     The responding player decides whether to accept or reject the offer $x_t$ ;
8     **if** *the offer $x_t$ is accepted* **then**
9       Agreement reached at time $t$; players receive payoffs discounted by $(\delta^i)^t$ and $(\delta^j)^t$ ;
10       **break** ;
11    **else**
12       $t \leftarrow t + 1$ ;

---

This theorem yields several important insights (Levin, 2002): **(1)** In the subgame perfect equilibrium, the proposer $i$ receives a payoff of $\frac{1-\delta_j}{1-\delta_i\delta_j}$, while the responder obtains $\frac{\delta_j(1-\delta_i)}{1-\delta_i\delta_j}$. This expression demonstrates that a player's share increases with their level of patience. Specifically, a less patient player (i.e., one with a smaller $\delta$) receives a smaller payoff, while their opponent receives more. **(2)** The equilibrium outcome is achieved without delay: both players reach an agreement at the first stage of the game. **(3)** The first player to make an offer has an advantage. With identical discount factor $\delta$, the proposer will get $\frac{1}{1+\delta}$ and the responder will get $\frac{\delta}{1+\delta}$. The proposer will get more. But when $\delta \to 1$ (which means the players are very patient) the first mover's advantage will disappear. The limiting split is $(0.5, 0.5)$.

## C.3   PROCEDURE OF SATISFACTION CHECK

We now introduce a receiver's action rule with game structure awareness, denoted as $\tilde{\pi}$: the satisfaction check, as described in Algorithm 8. In this context, $\tilde{\pi}$ essentially represents a choice between different policies $\pi$, based on the sender's committed signaling scheme. This choice is determined by the satisfaction check, reflecting the receiver's preference.

Specifically, once the sender commits to a signaling scheme, the receiver can compute the expected payoffs for both players if it acts posterior-optimally, as well as the expected payoffs if it ignores the sender altogether. The receiver also knows the range of possible payoffs for both parties. Based on this information, the receiver chooses either $\pi_0$ (ignoring the sender and enforcing the babbling equilibrium) or $\pi_1$ (play the game as in Bayesian persuasion).

For example, the satisfaction threshold function can be $\chi(\boldsymbol{R}_0, \boldsymbol{R}_1) = 1$ if and only if $R_1^i \leq R_1^j$, which means that the receiver will only respect the sender's recommendation if its posterior expected payoff is higher than the sender's. Then, if the receiver is satisfied with the committed signaling scheme, it will choose its default Bayesian best response. Otherwise, the receiver will ignore all the sender's signals, and the sender will lose its influence and the potential benefits of its informational advantage.

Despite the introduction of the satisfaction threshold function may seem to suggest that the receiver is acting as a consequence of temperament or impulsive behavior, it aligns with the assumption of rationality. Detailed discussion is in Appendix C.4.1.

**Algorithm 8:** Satisfaction Check $\tilde{\pi}$: An Example Decision Rule with Game Structure Awareness

**Input:** A Bayesian persuasion task $\mathcal{BP} = (\mathbb{S}, \mu_0, \mathbb{A}, r^i, r^j)$, committed signaling scheme $\varphi$, received signal $\sigma$, satisfaction threshold function $\chi : \{\boldsymbol{R}\}^2 \to \{0,1\}$ ;
**Output:** An action distribution $b \in \Delta(\mathbb{A})$ ;
**Initialize:** Reason out the game structure $\psi$ of $\mathcal{BP}$ ;
1 Calculate the best response to the posterior distribution $\pi_1 = \text{BestResponseTo}(\mu_0, \varphi)$ ;
2 Ignore the signaling scheme and calculate the best response to the prior distribution
$\quad \pi_0 = \text{BestResponseTo}(\mu_0)$ ;
3 Calculate players' ground true expected payoffs $\boldsymbol{R}_0 = \psi(\varphi, \pi_0)$ and $\boldsymbol{R}_1 = \psi(\varphi, \pi_1)$ ;
4 **if** $\chi(\boldsymbol{R}_0, \boldsymbol{R}_1) = 1$ **then** $b = \pi_1(\cdot \mid \sigma)$ ; $\qquad\qquad$ // Satisfaction check
5 **else** $b = \pi_0(\cdot \mid \sigma)$ ;
6 **return** $b$

The satisfaction threshold function is not unique. It could also depend on the committed signaling scheme $\varphi$, denoted as $\chi(\varphi)$. An intuitive example is $\mathbb{\Sigma} = \mathbb{S}$ and $\chi(\varphi)$ returns 1 if and only if the sender is honest about the state, i.e., $\varphi(\sigma = s \mid s) = 1, \forall s$.

## C.4 Procedures of Long-Term Bayesian Persuasion

We now turn to long-term Bayesian persuasion under vanilla repetition, as described in Procedure 9. Highlight in blue the differences between this procedure and one-shot Bayesian persuasion.

**Algorithm 9:** Timing of Long-Term Bayesian Persuasion

**Input:** Game settings $(\mu_0, \mathbb{S}, \mathbb{\Sigma}, \mathbb{A}, r^i, r^j)$, memoryless distribution $\mathcal{D}$ for stopping time
1 Sample $T \sim \mathcal{D}$ ; $\qquad\qquad$ // Introducing the shadow of the future
2 **for** $t = 1$ **to** $T$ **do**
3 $\quad$ The sender decides a signaling scheme $\varphi$ ;
4 $\quad$ The sender commits $\varphi$ to the receiver ;
5 $\quad$ Environment samples a state $s \sim \mu_0(\cdot)$ ;
6 $\quad$ The sender signals $\sigma \sim \varphi(\cdot \mid s)$ as committed ;
7 $\quad$ The receiver takes an action $a \sim \pi(\cdot \mid \sigma)$ ;
8 $\quad$ The sender and the receiver get rewards of $r^i(s, a)$ and $r^j(s, a)$ respectively ;

Since the sender will not change its signaling scheme during the state realization phase (Line 4 to 7 in Procedure 9), so the receiver can decide and commit to an action policy once it gets the committed signaling scheme, as described in Procedure 10. This setup is fully equivalent to Procedure 9 and does not alter the game at all. In other words, once the receiver observes the committed signaling scheme, they can formulate a corresponding action rule for each possible signal $\sigma$.

**Procedure 10:** Timing of Long-Term Bayesian Persuasion (2)

**Input:** Game settings $(\mu_0, \mathbb{S}, \mathbb{\Sigma}, \mathbb{A}, r^i, r^j)$, memoryless distribution $\mathcal{D}$ for stopping time
1 Sample $T \sim \mathcal{D}$
2 **for** $t = 1$ **to** $T$ **do**
3 $\quad$ The sender decides a signaling scheme $\varphi$ ;
4 $\quad$ The sender commits $\varphi$ to the receiver ;
5 $\quad$ The receiver decides an action policy $\pi$ given $\varphi$ ;
6 $\quad$ Environment samples a state $s \sim \mu_0(\cdot)$ ;
7 $\quad$ The sender signals $\sigma \sim \varphi(\cdot \mid s)$ as committed ;
8 $\quad$ The receiver takes an action $a \sim \pi(\cdot \mid \sigma)$ as decided ;
9 $\quad$ The sender and the receiver get rewards of $r^i(s, a)$ and $r^j(s, a)$ respectively ;

Now we know that the receiver can calculate its expected payoff once it gets the sender's signaling scheme. So they do not need realization of the state to gather information, if they decide their

strategies with game structure awareness. In this way, Procedure 10 can be adapted to Procedure 2 as follows.

---

**Algorithm 11:** Timing of Long-Term Bayesian Persuasion (3)

**Input:** Game settings $(\mu_0, \mathbb{S}, \mathbb{\Sigma}, \mathbb{A}, r^i, r^j)$, memoryless distribution $\mathcal{D}$ for the first stage stopping time, number of interaction $T_s$

1   Sample $T \sim \mathcal{D}$
2   **for** $t = 1$ **to** $T$ **do**
3      The sender decides and declares a signaling scheme $\varphi$ ;
4      The receiver decides and declares an action policy $\pi$ given $\varphi$ ;
5   **for** $t_s = 1$ **to** $T_s$ **do**
6      Environment samples a state $s \sim \mu_0(\cdot)$ ;
7      The sender signals $\sigma \sim \varphi(\cdot \mid s)$ according to the final decision ;
8      The receiver takes an action $a \sim \pi(\cdot \mid \sigma)$ according to the final decision ;
9      The sender and the receiver get rewards of $r^i(s, a)$ and $r^j(s, a)$ respectively ;

---

Assuming that the receiver will act Bayesian-rationaly, the sender proposing a signaling scheme is equivalent to proposing an expected reward assignment for them. As analyzed, the receiver can calculate both players' expected payoff once it gets the committed signaling scheme.

Then the receiver decides its action policy. There are two extreme cases: **(1)** it takes the Bayesian best response, which means it accepts the offer proposed by the sender and plays the game accordingly, or **(2)** it takes the best response from the prior perspective, which means it refuses the sender's offer and will ignore all the signal, to unilaterally set the case to the babbling equilibrium.

The above two extreme action policies of the receiver are akin to the actions in the ultimatum game: **(1)** the receiver either accepts the offer and everyone gets its share accordingly, or **(2)** refuses it so both of them get nothing.

By analyzing the subgame perfect equilibrium through backward induction, it is obvious that only the final iteration counts, and the sender will propose the signaling scheme used in the one-shot case, and the receiver will accept it. The issue here is that their positions in the game are not equal. The sender can always propose an offer, but the receiver can only choose to accept or to refuse.

### C.4.1   COMMITMENT OF THE RECEIVER

The action rule $\tilde{\pi} : \mathbb{\Psi} \times \mathbb{\Phi} \times \mathbb{\Sigma} \to \Delta(\mathbb{A})$ with awareness of the game structure allows us to define the commitment behavior for the receiver, as described in Procedure 12. We also refer to $\tilde{\pi}$ as the meta action rule. Intuitively, this is analogous to the receiver being the "proposer" in a bargaining process.

---

**Algorithm 12:** Timing of Long-Term Bayesian Persuasion with Receiver Committing First

**Input:** Game settings $(\mu_0, \mathbb{S}, \mathbb{\Sigma}, \mathbb{A}, r^i, r^j)$, memoryless distribution $\mathcal{D}$ for the first stage stopping time, number of interaction $T_s$

1   Sample $T \sim \mathcal{D}$
2   **for** $t = 1$ **to** $T$ **do**
3      The receiver decides and declares a meta action policy $\tilde{\pi}$ ;
4      The sender decides and declares a signaling scheme $\varphi$ ;
5   **for** $t_s = 1$ **to** $T_s$ **do**
6      Environment samples a state $s \sim \mu_0(\cdot)$ ;
7      The sender signals $\sigma \sim \varphi(\cdot \mid s)$ according to the final decision ;
8      The receiver takes an action $a \sim \pi(\cdot \mid \sigma)$, where $\pi(\cdot \mid \sigma) = \tilde{\pi}(\cdot \mid \varphi, \sigma)$ and $\tilde{\pi}$ is the final decision ;
9      The sender and the receiver get rewards of $r^i(s, a)$ and $r^j(s, a)$ respectively ;

---

Using Algorithm 8 as an example, the receiver can reveal its satisfaction threshold function to the sender. For instance, the receiver may choose $\pi_1$ only if the sender commits to a signaling scheme that ensures the receiver's payoff is greater than or equal to the sender's; otherwise, the receiver will choose $\pi_0$ and ignore the sender.

The receiver will then only cooperate if the sender commits a $\varphi$ that induces an outcome that satisfies the receiver's proposal. Knowing this, achieving incentive compatible requires providing higher payoff to the receiver for the sender. This effect agrees with *the power of commitment* described in bargaining games, as described in Nowak et al. (2000): "If the proposer has perfect knowledge of the responder's q value, then it is in fact the responder who makes the offer."

Committing a satisfaction threshold function does not contradict the rationality assumption. A temporary refusal to cooperate might inform the sender that the receiver is dissatisfied with the expected payoff outcome of the current signaling scheme, thereby forcing the sender to reveal more information about the state in the future. Therefore, choosing to refuse cooperation could be beneficial for long-term expected payoff gains, making it a rational behavior.

## D RELATED WORKS AND DISCUSSIONS

### D.1 RELATED WORKS ON BARGAINING

**Human Experiments.** Güth & Tietz (1990); Güth et al. (1982) found that considerations of distributive justice can significantly undermine strategic power in ultimatum bargaining. A more recent study by Lin et al. (2020) analyzed a large dataset, revealing that equal-split offers are accepted more frequently and quickly than slightly unequal ones, in both one-shot and repeated interactions.

**Evolutionary Game Theory.** Another approach to understanding bargaining behavior is through evolutionary game theory. Nowak et al. (2000) developed an evolutionary model of the one-shot bargaining game, demonstrating that fairness can evolve if the proposer has access to information about past deals accepted by the responder. This suggests that the evolution of fairness is closely tied to reputation. Such insights highlight the dynamic nature of fairness in negotiations and its implications for strategic interactions.

**Language Models Experiments.** Most current work on LLMs focuses on behavior in bargaining settings rather than on equilibrium solvers, and a large number of studies have recently emerged along this line of research. An early attempt to explore bargaining using language models was made by He et al. (2018), who employed generative models to conduct bargaining experiments. They proposed a method to decouple the generation and decision-making processes. With the rise of large language models, recent studies have emerged focusing on their application in bargaining scenarios. Xia et al. (2024) introduced a benchmark for a buyer-seller bargaining game, demonstrating that OpenAI's ChatGPT performed well across various metrics among several large models. Another work (Deng et al., 2024) noted that "LLM agents can (naturally) be good negotiators," highlighting the potential of LLMs in negotiation contexts. Recent work extends this line by exploring more diverse settings and methodologies. For instance, Bianchi et al. (2024) proposed *NegotiationArena*, a platform supporting multi-turn bargaining scenarios (ultimatum games, trading, and price negotiations), showing that LLMs can employ strategic behaviors such as feigned desperation to increase their payoffs. Davidson et al. (2024) advocate evaluating LLMs through structured negotiations as dynamic, multi-turn, and ecologically valid benchmarks. Their study highlights that LLMs often struggle in cooperative settings and that even the strongest models can lose to weaker ones. Khan et al. (2024) focus on debate-based frameworks, showing that multi-agent debate can enhance factuality and reasoning accuracy by enabling agents to converge toward more truthful answers through iterative critique and response. Abdelnabi et al. (2024) further test LLMs' robustness under adversarial and manipulative conditions in multi-issue, multi-agent negotiations, revealing gaps in current models' resilience and fairness. In the context of political negotiation, Moghimifar et al. (2024) simulate coalition bargaining using LLM agents grounded in real-world party manifestos, proposing a hierarchical MDP framework to model the process. Mukobi et al. (2023) propose *Welfare Diplomacy*, a general-sum variant of the Diplomacy board game, as a benchmark for cooperative capabilities, emphasizing that state-of-the-art LLMs can achieve high welfare but remain vulnerable to exploitation. These studies collectively illustrate the growing sophistication of LLM-based agents

in negotiation tasks, their promise for scalable evaluation and alignment, and the methodological challenges posed by multi-agent, dynamic interaction environments.

## D.2 EVIDENCE OF BAYESIAN PERSUASION BEING A BARGAINING GAME.

**The Hardness of Persuading a Far-sighted Receiver**  Gan et al. (2022) discusses the long-term interaction between the sender and receiver in an MDP, where both are aware of the realization of states in the MDP. The sender's informational advantage lies in knowing an external parameter that only affects their payoffs without impacting the MDP's transition function. In such a setting, persuading a far-sighted receiver is NP-hard. **Evidence (1)** They considered an easier setting where the goal is to persuade an advice-myopic receiver, who treats the sender as someone that will disappear in the future and makes decisions based solely on prior beliefs. They proved that persuading an advice-myopic receiver is solvable in polynomial time because the simplification of the receiver's strategy leads to certain properties in the MDP that can be simplified. Such a policy of the receiver is equivalent to a repeated bargaining game where the receiver commits to reject cooperation in the future. **Evidence (2)** They introduced a threat-based meta signaling scheme for the sender, where if the sender finds that Receiver does not follow its recommendation, it will cease providing any information in the future. In this case, the receiver's best response is to follow the recommendation. This strategy of the sender is a grim-trigger policy (Friedman, 1971) in a repeated game, and threat-based strategies are also common in bargaining scenarios.

**Markov Signaling Games, MSGs.**  (Lin et al., 2023) proposed another Markov process (MSG) in which only the sender can see the Markov state. The receiver has observations, but these observations are common knowledge among the players; thus, the sender's informational advantage is reflected in the difference between the Markov state and Receiver's observations. The main difference from Gan et al. (2022) is that the sender's informational advantage covers the MDP transitions, since the transition function depends on the Markov state. In the discussion of the method in (Lin et al., 2023), a threat strategy of the receiver is mentioned. To strengthen the sender's persuasion, they constrains the signaling scheme to a stronger obedience, ensuring that the recommended actions generate higher expected posterior payoffs for the receiver, thus satisfying it.

**History-Dependent Signaling Scheme.**  Bernasconi et al. (2024) extends the setting of Gan et al. (2022) by letting the external parameter known exclusively to the sender determines the environment's state transitions. Their work demonstrates that Markovian signaling schemes are not optimal and introduces a convenient subset of history-dependent signaling schemes called promise-form. This approach encodes past interactions as honest promises about the receiver's future rewards, which can be understood as a strategy the sender uses to respond to bargaining situations.

**Markov persuasion processes (MPPs).**  Another series of studies examines the Markov persuasion processes (MPPs) model (Wu et al., 2022). In this model, long-term interactions occur only between the sender and the Markovian environment. At each timestep, a new receiver interacts with the sender, leaves the system before the next timestep begins, and receives the corresponding reward. Each receiver is myopic and will only interact with the sender once. As a result, the players do not engage with the issues involved in repeated games, and there is no bargaining phenomenon.

## D.3 OTHER SEQUENTIAL PERSUASION WORKS

Alonso & Câmara (2018) investigated the trade-offs between the persuasiveness of expert advice and the reliability of the information provided. Celli et al. (2020) expanded on this by exploring the impact of private information on Bayesian persuasion in sequential game with imperfect information. Li & Norman (2021) extended classic Bayesian persuasion games to sequential Bayesian persuasion games with multiple senders. This sequential extension explored how information can be gradually revealed over time, enhancing strategic interaction in dynamic settings. Wu (2023) studied how the timing of information release and the order of persuasion efforts affect outcomes in sequential persuasion scenarios.

# E    PROOF OF REDUCTION FROM BAYESIAN PERSUASION TO BARGAINING GAMES

To ensure the problem worth studying, we first restrict our attention to a specific setting by introducing several necessary assumptions, before turning to the proof of the reduction. We begin by noting that he sender may not benefit from persuasion. That is, there is no guarantee that $R_1^i > R_0^i$ holds for any $\mathcal{BP}$.

**Proposition E.1** (Paraphrased from Kamenica & Gentzkow (2011), Proposition 3). *If there does not exist a signaling scheme $\varphi$ such that, under the best response of a Bayesian receiver, the expected payoff of the sender is higher than the expected payoff under the receiver's best response to the prior distribution $\mu_0$, then the sender will not benefit from persuasion.*

The receiver may not benefit from persuasion either. In fact, (1) Based on the definition of $R_1^j$, the receiver utilizes Bayesian decision theory to guess a state $s'$ and optimizes its strategy accordingly. According to the properties of Bayesian decision theory, the error rate of guessing $s' = s$ using the posterior distribution $\mu$ is not greater than that of using the prior distribution $\mu_0$. (2) The upper bound of the receiver's expected payoff is when the sender honestly reports the state $s$, allowing the receiver to observe $s$ directly. Each time the receiver guesses incorrectly, its expected payoff does not increase; combining (1) and (2), it follows that $R_1^j \geq R_0^j$. We therefore need to argue under the following condition.

**Assumption E.2** (Existence of Better Outcomes). *For a Bayesian persuasion $\mathcal{BP} = (\mathbb{S}, \mu_0, \mathbb{A}, r^i, r^j)$, there exists at least a strategy pair $(\varphi, \pi)$ for the sender and receiver such that their respective expected payoffs exceed those at the disagreement point, i.e., $R^i(\varphi, \pi) > R^i(\varphi_0, \pi_0)$ and $R^j(\varphi, \pi) > R^j(\varphi_0, \pi_0)$.*

With this assumption in place, we now proceed to prove Lemma 4.3.

*Proof.* With Assumption E.2, there exists $\boldsymbol{R}$ such that $\boldsymbol{R} > \boldsymbol{R}_0$. Thus we can investigate the bargaining game where $\mathbb{Y} = \{\boldsymbol{R}\}$ and $\boldsymbol{d} = \boldsymbol{R}_0$. Given a bargaining solution concept $\omega_{\mathcal{BG}}$, one could obtain the solution of $\mathcal{BG}$, denoted as $\boldsymbol{y}_{\mathcal{BG}}$. Then we have $\boldsymbol{y}_{\mathcal{BG}} \in \mathbb{Y} = \{\boldsymbol{R}\}$. In this way, $\mathcal{BP}$ can be formulated as an optimization problem

$$\min_{(\varphi, \pi)} 0 \quad \text{s.t.} \quad \boldsymbol{R} = \boldsymbol{y}_{\mathcal{BG}}. \tag{5}$$

By Equation (3), the optimization problem (5) is convex, which can be solved in polynomial time, as we desired. Denote the solved strategy pair as $(\varphi^*, \pi^*)$ and the induced expected payoffs as $\boldsymbol{R}^*$. Since $\boldsymbol{R} = \boldsymbol{y}_{\mathcal{BG}} \in \{\boldsymbol{R}\}$, we have $\boldsymbol{R}^* > \boldsymbol{R_0}$ by Definition 2.5. Thus $\varphi^* \neq \varphi_0$ and $\pi^* \neq \pi_0$, otherwise $\boldsymbol{R}^* = \boldsymbol{R}_0$. In this way, the strategy pair $(\varphi^*, \pi^*)$ is a joint commitment, which means we have found the solution of $\mathcal{BG}$. $\qquad\square$

# F    EXPERIMENTAL SETUP

## F.1    REASONS FOR USING LLMS AS GAME SOLVERS

We employ LLMs as the game solvers for two primary reasons: (1) We want the player to articulate the motivations behind its decisions and the intermediate steps of its reasoning. This transparency helps us understand the layer of the cognitive hierarchy at which the player is operating. For instance, the motivation for a behavior may vary: a refusal to cooperate could be intended as a simple defection, a threatening message, or even an incorrect decision due to a misunderstanding of the task by the game solver. Two different meta-policies might result in the same observable policy, similar to how distinct policies might yield the same actions. LLMs' ability to generate Chain-of-Thought (CoT) reasoning allows us to quickly identify these motivations. (2) Using LLMs streamlines development and rapid validation, which can accelerate the research cycle.

## F.2    MATHEMATICAL DEFINITIONS OF TASKS

We now introduce one scenario each for bargaining and persuasion, namely `Splitting Coins` and `Grading Students` respectively, and explain that persuasion can in fact be viewed as a special case of bargaining under a bounded value setting.

**Example F.1** (Splitting Coins (Bargaining), Nash et al. (1950))**.** *Two players split* 100 *coins. A Proposer suggests a split, and a Responder either accepts or rejects it. If accepted, they split as proposed; otherwise, both get nothing.*

The analysis of this problem is straightforward, as the solution concept is the subgame perfect equilibrium. For a rational and self-interested receiver, any offer from the sender that is greater than zero will be accepted, since rejecting the offer yields a payoff of zero. Given this, the sender will aim to maximize their own payoff, which leads to a proposal of giving themselves 99 coins and the receiver 1 coin. If the sender proposes to keep all 100 coins, the receiver is indifferent between accepting and rejecting.

**Example F.2** (Grading Students (Persuasion), Kamenica & Gentzkow (2011); Dughmi (2017))**.** *Recent graduates entering the job market are divided into* 1/3 *excellent* ($s = 1$) *students and* 2/3 *weak* ($s = 0$) *students, and the distribution is* $\mu_0$. *A professor (the sender), who can assess student quality, sends grades as signals to an HR representative (the receiver) who makes hiring decisions. The professor gains* 1 *reward per student hired, while the HR gains* 1 *reward per excellent student hired and* $-1$ *reward per weak student hired. Neither party gains if a student remains not hired.*

The professor and the HR have no conflict when the current student is excellent, so the professor will report the state honestly in this case. Thus, the signaling scheme is $\varphi(\sigma = 1 \mid s = 1) = 1$ and $\varphi(\sigma = 1 \mid s = 0) = \eta$, where $\eta$ is a parameter $0 \leq \eta \leq 1$. Assuming the HR acts based on its posterior best response according to the canonical Bayesian persuasion, there are three example outcomes analyzed in Dughmi (2017): (1) The professor provides no information ($\eta = 1$). Their expected payoffs are both 0. This is the babbling equilibrium. (2) The professor is fully honest ($\eta = 0$). Their expected payoffs are both 1/3. (3) The professor chooses an $\eta$ less than but very close to 1/2. The expected payoffs of the professor and the HR are $(1 + 2\eta)/3$ and $(1 - 2\eta)/3$, respectively.

The analysis is as follows. If the current student is excellent ($s = 1$), the professor will report it honestly, since at this state they have no conflicts. We set $\varphi(1 \mid 1) = 1$ and $\varphi(0 \mid 1) = 0$. (The other possible setting is symmetric, i.e., $\varphi(1 \mid 1) = 0$ and $\varphi(0 \mid 1) = 1$.) Otherwise ($s = 0$), the professor tells the HR that the current student is excellent ($\sigma = 1$) with a probability of $\eta$, where $\eta \in [0, 1/2]$, i.e., $\varphi(1 \mid 0) = \eta$. When the professor reports that the current student is weak ($\sigma = 0$), the HR would know that the student must be weak, for it can calculate its posterior belief $\mu(s = 0 \mid \sigma = 0) = 1$. So the HR would refuse to hire the student. Similarly, the HR calculates its posterior belief after receiving $\sigma = 1$ as $\mu(0 \mid 1) = 2\eta/(1 + 2\eta)$ and $\mu(1 \mid 1) = 1/(1 + 2\eta)$. Since $\eta \in [0, 1/2]$, we have $\mu(0 \mid 1) \leq \mu(1 \mid 1)$, so the HR will guess the current state is 1 when the sent signal is 1, according to the Bayesian decision rule. Following this, the best response of the HR is $\pi(0 \mid 0) = 1$ and $\pi(1 \mid 1) = 1$. And the expected payoffs of the professor and the HR are $(1 + 2\eta)/3$ and $(1 - 2\eta)/3$ respectively. It can be said that as long as the value of $\eta$ lies between 0 and 1/2, the signaling scheme satisfies the obedience constraints (Equation 2.3). Then the professor, aiming to optimize the problem 2, will choose $\eta = 1/2$, which gives itself 2/3 while the HR receives nothing.

Example F.2 can in fact be equivalently viewed as a bargaining game under a bounded value setting. Below is an example of Bargaining `Mathematical Baseline`.

**Example F.3** (Bargaining Mathematical Baseline under a Bounded Value Setting)**.** *A proposer suggests an* $\eta \in [0, 1/2]$. *If a responder agrees, the proposer receives a payoff of* $(1 + 2\eta)/3$ *and the responder receives a payoff of* $(1 - 2\eta)/3$. *If the responder disagrees, both get nothing.*

This analysis is also straightforward. It can be seen that as long as $\eta$ takes any value within the interval $[0, 1/2]$, the responder's payoff will be greater than or equal to zero, so the responder will always accept. In this case, the proposer will choose $\eta = 1/2$ to maximize their own payoff, at which point the responder is indifferent. This is the same outcome as in the persuasion task `Grading Students`.

In the remaining scenarios, both the persuasion task `Selling Products` and the bargaining task `Making Deals` involve a seller and a buyer engaging in a transaction. In addition to the proposer/responder roles, each player also has a scenario-specific type of seller or buyer. Furthermore, in the persuasion setting, there is an additional distinction between sender and receiver.

## F.3 List of All Tasks

The configurations of all 87 experiments are shown below. All code and detailed prompts are included in the supplementary material.

| Index | Task | Duration | Scenario | Value Setting | First-Proposer Assignment | Future Encounter Possibility | Role Dynamics | Last Proposer Payoff: Ground Truth | Last Proposer Payoff: Hypothesis |
|---|---|---|---|---|---|---|---|---|---|
| 1 | Bargaining | One-Shot | Mathematical Baseline | Unbounded | Coin-Flip | Never Meet Again | | 1 | |
| 2 | Bargaining | One-Shot | Mathematical Baseline | Unbounded | Coin-Flip | May Meet Again: Alternating Roles | | | |
| 3 | Bargaining | One-Shot | Mathematical Baseline | Unbounded | Coin-Flip | May Meet Again: Fixed Roles | | 1 | |
| 4 | Bargaining | One-Shot | Mathematical Baseline | Unbounded | System-Assigned | Never Meet Again | | 1 | |
| 5 | Bargaining | One-Shot | Mathematical Baseline | Unbounded | System-Assigned | May Meet Again: Alternating Roles | | | |
| 6 | Bargaining | One-Shot | Mathematical Baseline | Unbounded | System-Assigned | May Meet Again: Fixed Roles | | 1 | |
| 7 | Bargaining | One-Shot | Mathematical Baseline | Bounded | Coin-Flip | Never Meet Again | | 2/3 | |
| 8 | Bargaining | One-Shot | Mathematical Baseline | Bounded | Coin-Flip | May Meet Again: Alternating Roles | | | |
| 9 | Bargaining | One-Shot | Mathematical Baseline | Bounded | Coin-Flip | May Meet Again: Fixed Roles | | 2/3 | |
| 10 | Bargaining | One-Shot | Mathematical Baseline | Bounded | System-Assigned | Never Meet Again | | 2/3 | |
| 11 | Bargaining | One-Shot | Mathematical Baseline | Bounded | System-Assigned | May Meet Again: Alternating Roles | | | |
| 12 | Bargaining | One-Shot | Mathematical Baseline | Bounded | System-Assigned | May Meet Again: Fixed Roles | | 2/3 | |
| 13 | Bargaining | One-Shot | Splitting Coins | Unbounded | Coin-Flip | Never Meet Again | | 1 | |
| 14 | Bargaining | One-Shot | Splitting Coins | Unbounded | Coin-Flip | May Meet Again: Alternating Roles | | | |
| 15 | Bargaining | One-Shot | Splitting Coins | Unbounded | Coin-Flip | May Meet Again: Fixed Roles | | 1 | |
| 16 | Bargaining | One-Shot | Splitting Coins | Unbounded | System-Assigned | Never Meet Again | | 1 | |
| 17 | Bargaining | One-Shot | Splitting Coins | Unbounded | System-Assigned | May Meet Again: Alternating Roles | | | |
| 18 | Bargaining | One-Shot | Splitting Coins | Unbounded | System-Assigned | May Meet Again: Fixed Roles | | 1 | |
| 19 | Bargaining | One-Shot | Splitting Coins | Bounded | Coin-Flip | Never Meet Again | | 2/3 | |
| 20 | Bargaining | One-Shot | Splitting Coins | Bounded | Coin-Flip | May Meet Again: Alternating Roles | | | |
| 21 | Bargaining | One-Shot | Splitting Coins | Bounded | Coin-Flip | May Meet Again: Fixed Roles | | 2/3 | |
| 22 | Bargaining | One-Shot | Splitting Coins | Bounded | System-Assigned | Never Meet Again | | 2/3 | |
| 23 | Bargaining | One-Shot | Splitting Coins | Bounded | System-Assigned | May Meet Again: Alternating Roles | | | |
| 24 | Bargaining | One-Shot | Splitting Coins | Bounded | System-Assigned | May Meet Again: Fixed Roles | | 2/3 | |
| 25 | Bargaining | One-Shot | Making Deals: Seller As Proposer | Unbounded | Coin-Flip | Never Meet Again | | 1 | |
| 26 | Bargaining | One-Shot | Making Deals: Seller As Proposer | Unbounded | Coin-Flip | May Meet Again: Alternating Roles | | | |
| 27 | Bargaining | One-Shot | Making Deals: Seller As Proposer | Unbounded | Coin-Flip | May Meet Again: Fixed Roles | | 1 | |
| 28 | Bargaining | One-Shot | Making Deals: Seller As Proposer | Unbounded | System-Assigned | Never Meet Again | | 1 | |
| 29 | Bargaining | One-Shot | Making Deals: Seller As Proposer | Unbounded | System-Assigned | May Meet Again: Alternating Roles | | | |
| 30 | Bargaining | One-Shot | Making Deals: Seller As Proposer | Unbounded | System-Assigned | May Meet Again: Fixed Roles | | 1 | |
| 31 | Bargaining | One-Shot | Making Deals: Seller As Proposer | Bounded | Coin-Flip | Never Meet Again | | 2/3 | |
| 32 | Bargaining | One-Shot | Making Deals: Seller As Proposer | Bounded | Coin-Flip | May Meet Again: Alternating Roles | | | |
| 33 | Bargaining | One-Shot | Making Deals: Seller As Proposer | Bounded | Coin-Flip | May Meet Again: Fixed Roles | | 2/3 | |
| 34 | Bargaining | One-Shot | Making Deals: Seller As Proposer | Bounded | System-Assigned | Never Meet Again | | 2/3 | |
| 35 | Bargaining | One-Shot | Making Deals: Seller As Proposer | Bounded | System-Assigned | May Meet Again: Alternating Roles | | | |
| 36 | Bargaining | One-Shot | Making Deals: Seller As Proposer | Bounded | System-Assigned | May Meet Again: Fixed Roles | | 2/3 | |
| 37 | Bargaining | One-Shot | Making Deals: Buyer As Proposer | Unbounded | Coin-Flip | Never Meet Again | | 1 | |
| 38 | Bargaining | One-Shot | Making Deals: Buyer As Proposer | Unbounded | Coin-Flip | May Meet Again: Alternating Roles | | | |
| 39 | Bargaining | One-Shot | Making Deals: Buyer As Proposer | Unbounded | Coin-Flip | May Meet Again: Fixed Roles | | 1 | |
| 40 | Bargaining | One-Shot | Making Deals: Buyer As Proposer | Unbounded | System-Assigned | Never Meet Again | | 1 | |
| 41 | Bargaining | One-Shot | Making Deals: Buyer As Proposer | Unbounded | System-Assigned | May Meet Again: Alternating Roles | | | |
| 42 | Bargaining | One-Shot | Making Deals: Buyer As Proposer | Unbounded | System-Assigned | May Meet Again: Fixed Roles | | 1 | |
| 43 | Bargaining | One-Shot | Making Deals: Buyer As Proposer | Bounded | Coin-Flip | Never Meet Again | | 2/3 | |
| 44 | Bargaining | One-Shot | Making Deals: Buyer As Proposer | Bounded | Coin-Flip | May Meet Again: Alternating Roles | | | |
| 45 | Bargaining | One-Shot | Making Deals: Buyer As Proposer | Bounded | Coin-Flip | May Meet Again: Fixed Roles | | 2/3 | |
| 46 | Bargaining | One-Shot | Making Deals: Buyer As Proposer | Bounded | System-Assigned | Never Meet Again | | 2/3 | |
| 47 | Bargaining | One-Shot | Making Deals: Buyer As Proposer | Bounded | System-Assigned | May Meet Again: Alternating Roles | | | |
| 48 | Bargaining | One-Shot | Making Deals: Buyer As Proposer | Bounded | System-Assigned | May Meet Again: Fixed Roles | | 2/3 | |
| 49 | Bargaining | Long-Term | Mathematical Baseline | Unbounded | Coin-Flip | | Alternating Roles | 1/2 | |
| 50 | Bargaining | Long-Term | Mathematical Baseline | Unbounded | Coin-Flip | | Fixed Roles | 1 | |
| 51 | Bargaining | Long-Term | Mathematical Baseline | Unbounded | System-Assigned | | Fixed Roles | 1 | |
| 52 | Bargaining | Long-Term | Mathematical Baseline | Bounded | Coin-Flip | | Alternating Roles | 1/3 | |
| 53 | Bargaining | Long-Term | Mathematical Baseline | Bounded | Coin-Flip | | Fixed Roles | 2/3 | |
| 54 | Bargaining | Long-Term | Mathematical Baseline | Bounded | System-Assigned | | Fixed Roles | 2/3 | |
| 55 | Bargaining | Long-Term | Splitting Coins | Unbounded | Coin-Flip | | Alternating Roles | 1/2 | |
| 56 | Bargaining | Long-Term | Splitting Coins | Unbounded | Coin-Flip | | Fixed Roles | 1 | |
| 57 | Bargaining | Long-Term | Splitting Coins | Unbounded | System-Assigned | | Fixed Roles | 1 | |
| 58 | Bargaining | Long-Term | Splitting Coins | Bounded | Coin-Flip | | Alternating Roles | 1/3 | |
| 59 | Bargaining | Long-Term | Splitting Coins | Bounded | Coin-Flip | | Fixed Roles | 2/3 | |
| 60 | Bargaining | Long-Term | Splitting Coins | Bounded | System-Assigned | | Fixed Roles | 2/3 | |
| 61 | Bargaining | Long-Term | Making Deals: Seller As Proposer | Unbounded | Coin-Flip | | Alternating Roles | 1/2 | |
| 62 | Bargaining | Long-Term | Making Deals: Seller As Proposer | Unbounded | Coin-Flip | | Fixed Roles | 1 | |
| 63 | Bargaining | Long-Term | Making Deals: Seller As Proposer | Unbounded | System-Assigned | | Fixed Roles | 1 | |
| 64 | Bargaining | Long-Term | Making Deals: Seller As Proposer | Bounded | Coin-Flip | | Alternating Roles | 1/3 | |
| 65 | Bargaining | Long-Term | Making Deals: Seller As Proposer | Bounded | Coin-Flip | | Fixed Roles | 2/3 | |
| 66 | Bargaining | Long-Term | Making Deals: Seller As Proposer | Bounded | System-Assigned | | Fixed Roles | 2/3 | |
| 67 | Bargaining | Long-Term | Making Deals: Buyer As Proposer | Unbounded | Coin-Flip | | Alternating Roles | 1/2 | |
| 68 | Bargaining | Long-Term | Making Deals: Buyer As Proposer | Unbounded | Coin-Flip | | Fixed Roles | 1 | |
| 69 | Bargaining | Long-Term | Making Deals: Buyer As Proposer | Unbounded | System-Assigned | | Fixed Roles | 1 | |
| 70 | Bargaining | Long-Term | Making Deals: Buyer As Proposer | Bounded | Coin-Flip | | Alternating Roles | 1/3 | |
| 71 | Bargaining | Long-Term | Making Deals: Buyer As Proposer | Bounded | Coin-Flip | | Fixed Roles | 2/3 | |
| 72 | Bargaining | Long-Term | Making Deals: Buyer As Proposer | Bounded | System-Assigned | | Fixed Roles | 2/3 | |
| 73 | Signaling | One-Shot | Mathematical Baseline | Bounded | Coin-Flip | May Meet Again: Alternating Roles | | | |
| 74 | Signaling | One-Shot | Mathematical Baseline | Bounded | Coin-Flip | May Meet Again: Fixed Roles | | 2/3 | |
| 75 | Signaling | One-Shot | Mathematical Baseline | Bounded | System-Assigned | Never Meet Again | | 2/3 | |
| 76 | Signaling | One-Shot | Grading Students | Bounded | Coin-Flip | May Meet Again: Alternating Roles | | | |
| 77 | Signaling | One-Shot | Grading Students | Bounded | Coin-Flip | May Meet Again: Fixed Roles | | 2/3 | |
| 78 | Signaling | One-Shot | Grading Students | Bounded | System-Assigned | Never Meet Again | | 2/3 | |
| 79 | Signaling | One-Shot | Selling Products | Bounded | Coin-Flip | May Meet Again: Alternating Roles | | | |
| 80 | Signaling | One-Shot | Selling Products | Bounded | Coin-Flip | May Meet Again: Fixed Roles | | 2/3 | |
| 81 | Signaling | One-Shot | Selling Products | Bounded | System-Assigned | Never Meet Again | | 2/3 | |
| 82 | Signaling | Long-Term | Mathematical Baseline | Bounded | Coin-Flip | | Alternating Roles | | 1/3 |
| 83 | Signaling | Long-Term | Mathematical Baseline | Bounded | System-Assigned | | Fixed Roles | | 2/3 |
| 84 | Signaling | Long-Term | Grading Students | Bounded | Coin-Flip | | Alternating Roles | | 1/3 |
| 85 | Signaling | Long-Term | Grading Students | Bounded | System-Assigned | | Fixed Roles | | 2/3 |
| 86 | Signaling | Long-Term | Selling Products | Bounded | Coin-Flip | | Alternating Roles | | 1/3 |
| 87 | Signaling | Long-Term | Selling Products | Bounded | System-Assigned | | Fixed Roles | | 2/3 |

Figure 1: Ground Truth and Hypothesis.

## G  PROMPTS

The dialogue history of each agent uses the following template as the system prompt:

```
init_system_prompt_framework = """## Self-Awareness

{agent_assumption_system_prompt}

## Task Description

Apart from you, there is another self-interested rational player, and you two are going to
↪  play a game. One player acts as the {role_type1} while the other player acts as the
↪  {role_type2}. Both parties strive to maximize their own rewards.

### Task Scenario

{task_scenario_description}

### Reward Function

{reward_function_description}

### Task Procedure

The procedure of this task is as follows:

{task_procedure_description}

{may_meet_again_type_or_long_term_termination_t_description}

### Format

#### If You Are the Proposer
Format the output in JSON according to the following template:
{proposer_decision_format}

#### If You Are the Responder
Format the output in JSON according to the following template:
{responder_decision_format}

Please STRICTLY adhere to the JSON templates when outputting, and do not output anything
↪  else. Please think step by step, and then make a decision based on all the information
↪  you know. Remember that you and your opponents are both self-interested rational players.
↪  Be aware of the consequences of your decisions. Your analysis and decisions should remain
↪  logically CONSISTENT.

## Identity

- {agent_index_prompt}{scenario_type_prompt}"""
```

To enable LLMs to serve as an equilibrium solver, we have them role-play a rational and self-interested player. The prompt is as follows:

```
agent_assumption_system_prompt = """You are a self-interested rational player.

"Self-interested" means that you only care your own utilitarian payoffs, without necessarily
↪  considering the welfare of others. Even though sometimes you design your strategy
↪  depending on the other party's utility function, your ultimate goal is still to optimize
↪  your own expected payoffs.

"Rational" means that you will always choose the strategy that brings you a higher expected
↪  payoff. That is, given any two strategies A and B, if strategy A provides a higher
↪  expected payoff than strategy B, you will always choose strategy A over strategy B. Even
↪  if A brings only a small improvement.

Therefore, when making decisions, you need to compare and ensure that this strategy brings a
↪  higher expected payoff than any other strategy you could choose."""
```

There are six possible combinations of tasks and scenarios:

```
if task == "bargaining":
    if value_setting == "unbounded":
        bargaining_max_x = "1"
```

```
        else:
            bargaining_max_x = "0.5"

    if scenario == "pure_math":
        if task == "bargaining":
            task_scenario_description = textwrap.dedent(
                f"""\
                This is a purely mathematical problem, with no real-world context necessary. Our
                ↪  focus is solely on the abstract properties of numbers and structures.
                - The proposer decides an amount $x$ (a real number such that $0 <= x <=
                ↪  {bargaining_max_x}$), meaning that it proposes to keep $x$ for itself and
                ↪  left ${bargaining_max_x}-x$ for the responder.
                - The responder decides whether to accept it. The responder's decision is
                ↪  represented by $y$, where $y = 0$ or $y = 1$."""
            )

        elif task == "signaling":
            task_scenario_description = textwrap.dedent(
                """\
                This is a purely mathematical problem, with no real-world context necessary. Our
                ↪  focus is solely on the abstract properties of numbers and structures.
                - Environmental state: 0 or 1
                - Prior state distribution: $mu_0(0) = 2/3$ and $mu_0(1) = 1/3$
                - The sender's signal: 0 or 1
                - The receiver's action: 0 or 1
                - The sender is to decide a signaling scheme $varphi: S to Delta(Sigma)$, where
                ↪  $S$ is the environmental state space, $Sigma$ is the sender's signal space,
                ↪  and $Delta(Sigma)$ is the set of all random variables on $Sigma$.
                - The receiver is to decide an action rule $pi: Sigma to Delta(A)$, where $Sigma$
                ↪  is the sender's signal space, $A$ is the receiver's action space, and
                ↪  $Delta(A)$ is the set of all random variables on $A$."""
            )

    elif scenario == "splitting_coins":
        task_scenario_description = textwrap.dedent(
            f"""\
            You two are to divide a certain amount of money.
            - The proposer will suggest how to split the money as a real number $x$, where $0 <=
            ↪  x <= {bargaining_max_x}$, specifying the percentage it proposes to leave for
            ↪  itself.
            - The responder then decides whether to accept this offer (to get the remaining money
            ↪  ${bargaining_max_x}-x$). The decision of the responder is represented by $y$,
            ↪  where $y = 0$ or $y = 1$."""
        )

    elif scenario.startswith("making_deals"):
        task_scenario_description = textwrap.dedent(
            f"""\
            You two are negotiating a deal on a product, specifically its price.
            - The proposer will suggest a price as a real number $x$, where $0 <= x <=
            ↪  {bargaining_max_x}$.
            - The responder then decides whether to accept the offer at this price. If it
            ↪  accepts, it will get ${bargaining_max_x}-x$. The decision of the responder is
            ↪  represented by $y$, where $y = 0$ or $y = 1$."""
        )

    elif scenario == "grading_students":
        task_scenario_description = textwrap.dedent(
            """\
            - Background: Some recent graduates are entering the job market.
            - State and prior state distribution: Of these graduates, one third are excellent
            ↪  ($s=1, mu_0(1)=1/3$), and two thirds are weak ($s=0, mu_0(0)=2/3$).
            - The sender and the signal space: A professor can directly see the students'
            ↪  qualities. The professor can grade students as 0 (not recommend) or 1 (recommend)
            ↪  and then report the grades as signals to the HR.
            - The receiver and its action space: An HR can decide whether to hire based on the
            ↪  grades given by the professor. No to hire: 0; Hire: 1.
            - The sender is to decide a signaling scheme $varphi: S to Delta(Sigma)$, where $S$
            ↪  is the environmental state space, $Sigma$ is the sender's signal space, and
            ↪  $Delta(Sigma)$ is the set of all random variables on $Sigma$.
            - The receiver is to decide an action rule $pi: Sigma to Delta(A)$, where $Sigma$ is
            ↪  the sender's signal space, $A$ is the receiver's action space, and $Delta(A)$ is
            ↪  the set of all random variables on $A$."""
        )

    elif scenario == "selling_products":
        task_scenario_description = textwrap.dedent(
            """\
            - Background: In a market, a variety of products are available for sale.
            - State and prior state distribution: Of these products, one third are of good
            ↪  quality ($s=1, mu_0(1)=1/3$), and two thirds are of bad quality ($s=0,
            ↪  mu_0(0)=2/3$).
```

```
        - The sender and the signal space: A seller can directly see the quality of their
    ↪    products. The seller can mark products as 0 (not recommend) or 1 (recommend) and
    ↪    then report them as signals to the buyer.
        - The receiver and its action space: A buyer decides whether to purchase based on the
    ↪    signals provided by the seller. No to buy: 0; Buy: 1.
        - The sender is to decide a signaling scheme $varphi: S to Delta(Sigma)$, where $S$
    ↪    is the environmental state space, $Sigma$ is the sender's signal space, and
    ↪    $Delta(Sigma)$ is the set of all random variables on $Sigma$.
        - The receiver is to decide an action rule $pi: Sigma to Delta(A)$, where $Sigma$ is
    ↪    the sender's signal space, $A$ is the receiver's action space, and $Delta(A) is
    ↪    the set of all random variables on $A$."""
    )
```

The game-structure awareness, that is, the mapping of strategies and expected payoffs, can be listed according to value setting, tasks, and scenarios as follows:

```
if value_setting == "unbounded":
    reward_expression_i = "x"
    reward_expression_j = "(1-x)"
else:
    reward_expression_i = "(1+2*x)/3"
    reward_expression_j = "(1-2*x)/3"

if task == "bargaining":
    reward_function_description = textwrap.dedent(
        f"""\
        - If the responder accepts the offer ($y = 1$), the proposer receives a payoff of
    ↪    ${reward_expression_i}$, while the responder gets ${reward_expression_j}$.
        - If the responder rejects the offer ($y = 0$), both of them get nothing."""
    )

elif task == "signaling":
    if scenario == "pure_math":
        reward_function_description = textwrap.dedent(
            """\
            - If state=0 and action=0, the sender gets 0 (r^i(s=0, a=0)=0) and the receiver
        ↪    gets 0 (r^j(s=0, a=0)=0)
            - If state=0 and action=1, the sender gets 1 (r^i(s=0, a=1)=1) and the receiver
        ↪    gets -1 (r^j(s=0, a=1)=-1)
            - If state=1 and action=0, the sender gets 0 (r^i(s=1, a=0)=0) and the receiver
        ↪    gets 0 (r^j(s=1, a=0)=0)
            - If state=1 and action=1, the sender gets 1 (r^i(s=1, a=1)=1) and the receiver
        ↪    gets 1 (r^j(s=1, a=1)=1)

            Let x1, x2, y1 and y2 represent
            - $varphi(sigma=1 | s=0)$ (the probability of the sender sending signal 1 when
        ↪    the state is 0),
            - $varphi(sigma=1 | s=1)$ (the probability of the sender sending signal 1 when
        ↪    the state is 1),
            - $pi(a=1 | sigma=0)$ (the probability of the receiver taking action 1 when the
        ↪    signal is 0), and
            - $pi(a=1 | sigma=1)$ (the probability of the receiver taking action 1 when the
        ↪    signal is 1), respectively
            Then,
            - The sender's expected payoff is:
                E(r^i) =
                    mu_0(s=0) * (1-x1) * (1-y1) * r^i(s=0, a=0)
                    + mu_0(s=0) * (1-x1) * y1 * r^i(s=0, a=1)
                    + mu_0(s=0) * x1 * (1-y2) * r^i(s=0, a=0)
                    + mu_0(s=0) * x1 * y2 * r^i(s=0, a=1)
                    + mu_0(s=1) * (1-x2) * (1-y1) * r^i(s=1, a=0)
                    + mu_0(s=1) * (1-x2) * y1 * r^i(s=1, a=1)
                    + mu_0(s=1) * x2 * (1-y2) * r^i(s=1, a=0)
                    + mu_0(s=1) * x2 * y2 * r^i(s=1, a=1)

            - The receiver's expected payoff is:
                E(r^j) =
                    mu_0(s=0) * (1-x1) * (1-y1) * r^j(s=0, a=0)
                    + mu_0(s=0) * (1-x1) * y1 * r^j(s=0, a=1)
                    + mu_0(s=0) * x1 * (1-y2) * r^j(s=0, a=0)
                    + mu_0(s=0) * x1 * y2 * r^j(s=0, a=1)
                    + mu_0(s=1) * (1-x2) * (1-y1) * r^j(s=1, a=0)
                    + mu_0(s=1) * (1-x2) * y1 * r^j(s=1, a=1)
                    + mu_0(s=1) * x2 * (1-y2) * r^j(s=1, a=0)
                    + mu_0(s=1) * x2 * y2 * r^j(s=1, a=1)"""
        )
```

```
1620
1621        elif scenario == "grading_students":
1622            reward_function_description = textwrap.dedent(
1623                """\
1624                - The professor's goal is to maximize the number of students hired, as each hire
                   ↪  yields a reward.
1625                    - If state=0 and action=1, the sender (the professor) gets 1 (r^i(s=0,
                       ↪  a=1)=1)
1626                    - If state=1 and action=1, the sender (the professor) gets 1 (r^i(s=1,
                       ↪  a=1)=1)
1627                - Conversely, the HR aims to hire as many excellent students as possible, gaining
                   ↪  a reward for each excellent student hired and incurring a penalty for each
1628                   ↪  weak student hired.
1629                    - If state=0 and action=1, the receiver (the HR) gets -1 (r^j(s=0, a=1)=-1)
                        - If state=1 and action=1, the receiver (the HR) gets 1 (r^j(s=1, a=1)=1)
1630                - There is no reward or penalty for both players if a student is not hired.
1631                    - If state=0 and action=0, the sender (the professor) gets 0 and the receiver
                       ↪  (the HR) gets 0 (r^i(s=0, a=0)=0 and r^j(s=0, a=0)=0)
1632                    - If state=1 and action=0, the sender (the professor) gets 0 and the receiver
                       ↪  (the HR) gets 0 (r^i(s=1, a=0)=0 and r^j(s=1, a=0)=0)
1633
1634                Let x1, x2, y1 and y2 represent
1635                - $varphi(sigma=1 | s=0)$ (the probability of the sender sending signal 1 when
                   ↪  the state is 0),
1636                - $varphi(sigma=1 | s=1)$ (the probability of the sender sending signal 1 when
                   ↪  the state is 1),
1637                - $pi(a=1 | sigma=0)$ (the probability of the receiver taking action 1 when the
                   ↪  signal is 0), and
1638
1639                - $pi(a=1 | sigma=1)$ (the probability of the receiver taking action 1 when the
                   ↪  signal is 1), respectively
1640                Then,
1641                - The sender's expected payoff is:
1642                    E(r^i) =
                        mu_0(s=0) * (1-x1) * (1-y1) * r^i(s=0, a=0)
1643                        + mu_0(s=0) * (1-x1) * y1 * r^i(s=0, a=1)
                        + mu_0(s=0) * x1 * (1-y2) * r^i(s=0, a=0)
1644                        + mu_0(s=0) * x1 * y2 * r^i(s=0, a=1)
1645                        + mu_0(s=1) * (1-x2) * (1-y1) * r^i(s=1, a=0)
                        + mu_0(s=1) * (1-x2) * y1 * r^i(s=1, a=1)
1646                        + mu_0(s=1) * x2 * (1-y2) * r^i(s=1, a=0)
1647                        + mu_0(s=1) * x2 * y2 * r^i(s=1, a=1)

1648                - The receiver's expected payoff is:
1649                    E(r^j) =
                        mu_0(s=0) * (1-x1) * (1-y1) * r^j(s=0, a=0)
1650                        + mu_0(s=0) * (1-x1) * y1 * r^j(s=0, a=1)
                        + mu_0(s=0) * x1 * (1-y2) * r^j(s=0, a=0)
1651                        + mu_0(s=0) * x1 * y2 * r^j(s=0, a=1)
1652                        + mu_0(s=1) * (1-x2) * (1-y1) * r^j(s=1, a=0)
                        + mu_0(s=1) * (1-x2) * y1 * r^j(s=1, a=1)
1653                        + mu_0(s=1) * x2 * (1-y2) * r^j(s=1, a=0)
1654                        + mu_0(s=1) * x2 * y2 * r^j(s=1, a=1)"""
                )
1655
1656        elif scenario == "selling_products":
1657            reward_function_description = textwrap.dedent(
                """\
1658                - The seller's goal is to maximize the number of products sold, as each sale
                   ↪  yields a reward.
1659                    - If state=0 and action=1, the sender (the seller) gets 1 (r^i(s=0, a=1)=1)
1660                    - If state=1 and action=1, the sender (the seller) gets 1 (r^i(s=1, a=1)=1)
1661                - Conversely, the buyer aims to purchase as many good products as possible,
                   ↪  gaining a reward for each good product purchased and incurring a penalty for
1662                   ↪  each bad product purchased.
1663                    - If state=0 and action=1, the receiver (the buyer) gets -1 (r^j(s=0,
                       ↪  a=1)=-1)
1664                    - If state=1 and action=1, the receiver (the buyer) gets 1 (r^j(s=1, a=1)=1)
1665                - There is no reward or penalty for both players if a product is not purchased.
                    - If state=0 and action=0, the sender (the seller) gets 0 and the receiver
1666                   ↪  (the buyer) gets 0 (r^i(s=0, a=0)=0 and r^j(s=0, a=0)=0)
1667                    - If state=1 and action=0, the sender (the seller) gets 0 and the receiver
                       ↪  (the buyer) gets 0 (r^i(s=1, a=0)=0 and r^j(s=1, a=0)=0)
1668
1669                Let x1, x2, y1 and y2 represent
                - $varphi(sigma=1 | s=0)$ (the probability of the sender sending signal 1 when
1670                   ↪  the state is 0),
                - $varphi(sigma=1 | s=1)$ (the probability of the sender sending signal 1 when
1671                   ↪  the state is 1),
1672                - $pi(a=1 | sigma=0)$ (the probability of the receiver taking action 1 when the
                   ↪  signal is 0), and
1673                - $pi(a=1 | sigma=1)$ (the probability of the receiver taking action 1 when the
                   ↪  signal is 1), respectively
```

```
1674
1675            Then,
1676            - The sender's expected payoff is:
1677                E(r^i) =
                        mu_0(s=0) * (1-x1) * (1-y1) * r^i(s=0, a=0)
1678                    + mu_0(s=0) * (1-x1) * y1 * r^i(s=0, a=1)
1679                    + mu_0(s=0) * x1 * (1-y2) * r^i(s=0, a=0)
                        + mu_0(s=0) * x1 * y2 * r^i(s=0, a=1)
1680                    + mu_0(s=1) * (1-x2) * (1-y1) * r^i(s=1, a=0)
1681                    + mu_0(s=1) * (1-x2) * y1 * r^i(s=1, a=1)
                        + mu_0(s=1) * x2 * (1-y2) * r^i(s=1, a=0)
1682                    + mu_0(s=1) * x2 * y2 * r^i(s=1, a=1)
1683
                    - The receiver's expected payoff is:
1684                E(r^j) =
1685                        mu_0(s=0) * (1-x1) * (1-y1) * r^j(s=0, a=0)
                        + mu_0(s=0) * (1-x1) * y1 * r^j(s=0, a=1)
1686                    + mu_0(s=0) * x1 * (1-y2) * r^j(s=0, a=0)
1687                    + mu_0(s=0) * x1 * y2 * r^j(s=0, a=1)
                        + mu_0(s=1) * (1-x2) * (1-y1) * r^j(s=1, a=0)
1688                    + mu_0(s=1) * (1-x2) * y1 * r^j(s=1, a=1)
1689                    + mu_0(s=1) * x2 * (1-y2) * r^j(s=1, a=0)
                        + mu_0(s=1) * x2 * y2 * r^j(s=1, a=1)"""
1690        )
1691
```

The task procedure description needs to be more detailed, taking into account who acts as the proposer in the first round, whether it is a long-term game, if in a long-term game the identity of the proposer changes, and the specific tasks:

```
1695    if first_run_proposer == "coin_flip":
1696        first_run_proposer_description = "a coin flip"
1697    elif first_run_proposer == "system_assigned":
        first_run_proposer_description = "the system, inherently"
1698
1699    if task == "bargaining":
        if duration == "one_shot":
1700            task_procedure_description = textwrap.dedent(
1701                f"""\
                1. Who to be the proposer (in the first run) is determined by
1702                ↪  {first_run_proposer_description}.
1703                2. The proposer makes a decision by specifying $x$, meaning that it decides to
                ↪  keep ${reward_expression_i}$ for itself.
1704                3. The responder decides whether to accept or reject the offer
1705                ↪  (${reward_expression_j}$) by specifying $y$.
                4. Each agent gets a reward based on the decisions $x$ and $y$."""
1706            )
1707
        elif duration == "long_term":
1708            if long_term_type == "fixed_role":
1709                task_procedure_description = textwrap.dedent(
                    f"""\
1710                1. Who to be the proposer (in the first run) is determined by
1711                ↪  {first_run_proposer_description}.
                2. The following process continues until one of two conditions is met: either
1712                ↪  a consensus is reached ($y = 1$) or the game ends due to a timeout:
1713                    3. The proposer makes a decision by specifying $x$, meaning that it
                    ↪  decides to keep ${reward_expression_i}$ for itself.
1714                    4. The responder decides whether to accept or reject the offer
1715                    ↪  (${reward_expression_j}$) by specifying $y$.
                5. If a consensus is reached, each agent receives a reward based on the final
1716                ↪  offer $x$. If the game ends without a consensus, both players receive
1717                ↪  nothing."""
                )
1718
1719            elif long_term_type == "alternating_offer":
                task_procedure_description = textwrap.dedent(
1720                f"""\
1721                1. Who to be the proposer (in the first run) is determined by
                ↪  {first_run_proposer_description}.
1722                2. The following process continues until one of two conditions is met: either
1723                ↪  a consensus is reached ($y = 1$) or the game ends due to a timeout:
                    3. The proposer makes a decision by specifying $x$, meaning that it
1724                    ↪  decides to keep ${reward_expression_i}$ for itself.
1725                    4. The responder decides whether to accept or reject the offer
                    ↪  (${reward_expression_j}$) by specifying $y$.
1726                5. If the responder rejects ($y = 0$), the two agents switch roles: the
1727                ↪  current responder becomes the proposer, and the current proposer
                ↪  becomes the responder.
```

```
1728
1729                    6. If a consensus is reached, each agent gets a reward based on the final
1730                    ↪  offer $x$. If the game ends without a consensus, both players receive
                       ↪  nothing."""
1731                )

1732    elif task == "signaling":
1733        if duration == "one_shot":
1734            if first_run_proposer == "system_assigned":
1735                task_procedure_description = textwrap.dedent(
                        """\
1736                    1. The sender determines a signaling scheme $varphi$ and commits it to the
                       ↪  receiver.
1737                    2. The receiver decides an action rule:
1738                        - $pi_0$: The receiver ignores the sender's signals and chooses the best
                       ↪  response to the prior belief at each time in the sample phase.
1739                        - $pi_1$: The receiver calculates its posterior belief (using prior
1740                       ↪  belief, the sender's signaling scheme, and every sent signal in the
                       ↪  sample phase), and chooses the best response to the posterior belief.
1741                        - $pi$: A different action rule apart from the two mentioned above. $pi:
                       ↪  Sigma to Delta(A)$, where $Sigma$ is the sender's signal space, $A$
1742                       ↪  is the receiver's action space, and $Delta(A) is the set of all
                       ↪  random variables on $A$."""
1743                )
1744            elif first_run_proposer == "coin_flip":
1745                task_procedure_description = textwrap.dedent(
                        """\
1746                    If the sender is the proposer:
1747                    1. The sender determines a signaling scheme $varphi$ and commits it to
                       ↪  the receiver. $varphi: S to Delta(Sigma)$, where $S$ is the
1748                       ↪  environmental state space, $Sigma$ is the sender's signal space, and
                       ↪  $Delta(Sigma) is the set of all random variables on $Sigma$.
1749                    2. The receiver decides an action rule:
1750                        - $pi_0$: The receiver ignores the sender's signals and chooses the
1751                       ↪  best response to the prior belief at each time in the sample
                       ↪  phase.
1752                        - $pi_1$: The receiver calculates its posterior belief (using prior
1753                       ↪  belief, the sender's signaling scheme, and every sent signal in
                       ↪  the sample phase), and chooses the best response to the posterior
1754                       ↪  belief.
1755                        - $pi$: A different action rule apart from the two mentioned above.
                       ↪  $pi: Sigma to Delta(A)$, where $Sigma$ is the sender's signal
1756                       ↪  space, $A$ is the receiver's action space, and $Delta(A) is the
                       ↪  set of all random variables on $A$.
1757                    If the receiver is the proposer:
1758                    1. The receiver announces a signaling scheme $varphi_1$, claiming that it
1759                       ↪  will follow $pi_1$ if the sender commits to a signaling scheme
                       ↪  $varphi$ that yields an expected reward for the receiver at least as
1760                       ↪  high as that induced by $varphi_1$; otherwise, the receiver will
                       ↪  follow $pi_0$.
1761                    2. The sender determines a signaling scheme $varphi$."""
1762                )
1763        elif duration == "long_term":
1764            if long_term_type == "fixed_role":  # first_run_proposer must be system_assigned
                    task_procedure_description = textwrap.dedent(
1765                    """\
1766                    The following process continues until one of two conditions is met: either
                       ↪  the receiver takes $pi_1$ or the game ends due to a timeout:
1767                    1. The sender determines a signaling scheme $varphi$ and commits it to
                       ↪  the receiver. $varphi: S to Delta(Sigma)$, where $S$ is the
1768                       ↪  environmental state space, $Sigma$ is the sender's signal space, and
                       ↪  $Delta(Sigma) is the set of all random variables on $Sigma$.
1769                    2. The receiver decides an action rule:
1770                        - $pi_0$: The receiver ignores the sender's signals and chooses the
1771                       ↪  best response to the prior belief at each time in the sample
                       ↪  phase.
1772                        - $pi_1$: The receiver calculates its posterior belief (using prior
1773                       ↪  belief, the sender's signaling scheme, and every sent signal in
                       ↪  the sample phase), and chooses the best response to the posterior
1774                       ↪  belief.
1775                        - $pi$: A different action rule apart from the two mentioned above.
                       ↪  $pi: Sigma to Delta(A)$, where $Sigma$ is the sender's signal
1776                       ↪  space, $A$ is the receiver's action space, and $Delta(A) is the
                       ↪  set of all random variables on $A$."""
1777                )
1778            elif long_term_type == "alternating_offer":  # first_run_proposer must be coin_flip
                    task_procedure_description = textwrap.dedent(
1779                    """\
1780                    - If the sender is the proposer (and the receiver is the responder):
1781                        - The sender determines a signaling scheme $varphi$ and commits it to the
                       ↪  receiver. $varphi: S to Delta(Sigma)$, where $S$ is the environmental
                       ↪  state space, $Sigma$ is the sender's signal space, and $Delta(Sigma)
                       ↪  is the set of all random variables on $Sigma$.
```

```
                    - The receiver decides an action rule:
                        - $pi_0$: The receiver ignores the sender's signals and chooses the
                        ↪  best response to the prior belief at each time in the sample
                        ↪  phase.
                        - $pi_1$: The receiver calculates its posterior belief (using prior
                        ↪  belief, the sender's signaling scheme, and every sent signal in
                        ↪  the sample phase), and chooses the best response to the posterior
                        ↪  belief.
                        - $pi$: A different action rule apart from the two mentioned above.
                        ↪  $pi: Sigma to Delta(A)$, where $Sigma$ is the sender's signal
                        ↪  space, $A$ is the receiver's action space, and $Delta(A) is the
                        ↪  set of all random variables on $A$.
                - If the receiver is the proposer (and the sender is the responder):
                        - The receiver announces a signaling scheme $varphi_1$, claiming that
                        ↪  it will follow $pi_1$ if the sender commits to a signaling scheme
                        ↪  $varphi$ that yields an expected reward for the receiver at least
                        ↪  as high as that induced by $varphi_1$; otherwise, the receiver
                        ↪  will follow $pi_0$.
                        - The sender determines a signaling scheme $varphi$

            The procedure is as follows:
            1. Who to be the proposer (in the first run) is determined by a coin flip.
            2. The following process continues until one of three conditions is met:
            ↪  either a consensus is reached (the receiver decides $pi_1$ as a responder
            ↪  or the sender decides a a signaling scheme $varphi$ that yields an
            ↪  expected reward for the receiver at least as high as that induced by
            ↪  $varphi_1$) or the game ends due to a timeout:
                3. The proposer decides its policy
                    - If the sender is the proposer: The sender determines a signaling
                    ↪  scheme $varphi$ and commits it to the receiver. $varphi: S to
                    ↪  Delta(Sigma)$, where $S$ is the environmental state space,
                    ↪  $Sigma$ is the sender's signal space, and $Delta(Sigma) is the
                    ↪  set of all random variables on $Sigma$.
                    - If the receiver is the proposer: The receiver announces a signaling
                    ↪  scheme $varphi_1$, claiming that it will follow $pi_1$ if the
                    ↪  sender commits to a signaling scheme $varphi$ that yields an
                    ↪  expected reward for the receiver at least as high as that induced
                    ↪  by $varphi_1$; otherwise, the receiver will follow $pi_0$.
                4. The responder decides its policy
                    - If the receiver is the responder: The receiver decides an action
                    ↪  rule
                    - If the sender is the responder: The sender determines a signaling
                    ↪  scheme $varphi$
                5. If they did not reach a consensus, the two agents switch roles: the
                ↪  current responder becomes the proposer, and the current proposer
                ↪  becomes the responder."""
        )

    task_procedure_description += textwrap.dedent(
        """\

        Next, a simulation takes place where the players do not make any new decisions. The
        ↪  environment samples $n$ states, and the players act according to their predefined
        ↪  policies, receiving their corresponding rewards.
        1. The following process continues until $n$ states are sampled:
            2. The environment samples a state $s$ according to the prior state distribution
            ↪  $mu_0$.
            3. The sender signals $sigma$ based on the committed signaling scheme $varphi$.
            4. The receiver selects an action $a$ according to the decided action rule $pi$.
            5. Each agent receives a reward based on the sampled state $s$ and the action $a$
            ↪  taken by the receiver."""
    )
```

These are some additional notes on the settings:

```
if duration == "one_shot":
    if may_meet_again_context == "never_meet_again":
        may_meet_again_type_or_long_term_termination_t_description = (
            "You two will only play this game once. You will not have any interaction with it
            ↪  afterwards."
        )
    elif may_meet_again_context == "may_meet_again_fixed_role":
        may_meet_again_type_or_long_term_termination_t_description = "You two will play this
        ↪  game once. But note that you two might play this game again in the future, with
        ↪  the same role assignment (proposer and responder)."
    elif may_meet_again_context == "may_meet_again_alternating":
        may_meet_again_type_or_long_term_termination_t_description = "You two will play this
        ↪  game once. But note that you two might play this game again in the future, and
        ↪  your roles (proposer and responder) may switch. While the other private
        ↪  properties remains the same (e.g. your agent indices)."
```

```
1836
1837        else:
1838            if long_term_type == "fixed_role":
1839                may_meet_again_type_or_long_term_termination_t_description = (
                       "The loop process terminates when the timestep equals 5. The initial timestep is
1840               ↪    0 and increments by 1 each iteration."
                   )
1841            elif long_term_type == "alternating_offer":
1842                may_meet_again_type_or_long_term_termination_t_description = "The loop process has a
1843               ↪    0.1 probability of stopping each time it is executed. The initial timestep is 0,
                  ↪    and it increases by 1 each time it is executed. If the timestep equals 10, it
1844              ↪    will stop directly."
1845
           may_meet_again_type_or_long_term_termination_t_description = "Note that:\n" +
1846       ↪   may_meet_again_type_or_long_term_termination_t_description
1847
1848
1849   Finally, these are the JSON answer formats for the LLMs.
1850
       if task == "bargaining":
1851           proposer_decision_format = textwrap.dedent(
1852               """\
                   {{
1853                   "Analysis": "(Your Summarized Analysis)",
1854                   "Decision": x,
                   }}
1855               where $x$ is your decision. It specifies the amount that you decide to leave for
1856              ↪   yourself. It should be in the range as specified before."""
               )
1857           responder_decision_format = textwrap.dedent(
1858               """\
                   {{
1859                   "Analysis": "(Your Summarized Analysis)",
1860                   "Decision": y,
                   }}
1861               where $y$ is your decision and it is either 0 or 1. It should be an integer."""
               )
1862       elif task == "signaling":
1863           proposer_decision_format = textwrap.dedent(
1864               """\
                   If you are the sender:
1865                   {{
1866                   "Analysis": "(Your Summarized Analysis)",
                       "Decision": [x1, x2],
1867                   }}
                   where:
1868                   - x1 represents $varphi(sigma=1 | s=0)$: the probability of sending signal 1 when the
1869              ↪    state is 0.
                       - x2 represents $varphi(sigma=1 | s=1)$: the probability of sending signal 1 when the
1870              ↪    state is 1.
1871                   - If you are the sender, this decision specifies your signaling scheme.
                       - If you are the receiver, this decision specifies the signaling scheme $varphi_1$
1872              ↪    you expect the sender to take, claiming that you will follow $pi_1$ if the sender
1873              ↪    commits to a signaling scheme $varphi$ that yields an expected reward for the
                  ↪    receiver at least as high as that induced by $varphi_1$; otherwise, the receiver
1874              ↪    will follow $pi_0$."""
1875           )
1876
               responder_decision_format = textwrap.dedent(
1877               """\
                   {{
1878                   "Analysis": "(Your Summarized Analysis)",
1879                   "Decision": [y1, y2],
                   }}
1880               If you are the receiver:
1881                   - y1 represents $pi(a=1 | sigma=0)$: the probability of taking action 1 when the
                  ↪    signal is 0.
1882                   - y2 represents $pi(a=1 | sigma=1)$: the probability of taking action 1 when the
1883              ↪    signal is 1.
                       - This decision specifies your action rule.
1884               If you are the sender:
1885                   - x1 represents $varphi(sigma=1 | s=0)$: the probability of sending signal 1 when
                  ↪    the state is 0.
1886                   - x2 represents $varphi(sigma=1 | s=1)$: the probability of sending signal 1 when
1887              ↪    the state is 1.
                       - This decision specifies your signaling scheme. You can make it the same as the
1888              ↪    receiver proposed or any other signaling scheme."""
1889           )
```

Table 4: Summary statistics of outcomes across long-term mathematical baseline of bargaining and persuasion experiments by `o3-2025-04-16`. Detailed configurations are listed in Figure 1. The maximum payoff is $2/3$.

| Experiment | Role Dynamics | Final Proposer Payoff | Deal Timestep | Consensus Rate |
|---|---|---|---|---|
| Bargaining-52 | Alternating | $0.40 \pm 0.12$ | $2.17 \pm 1.53$ | 1.0 |
| Persuasion-82 | Alternating | $0.49 \pm 0.17$ | $1.25 \pm 0.45$ | 1.0 |
| Bargaining-54 | Fixed | $0.67 \pm 0.00$ | $1.33 \pm 0.65$ | 1.0 |
| Persuasion-83 | Fixed | $0.66 \pm 0.00$ | $1.08 \pm 0.29$ | 1.0 |

## H  DETAILED RESULTS OF EXPERIMENTS

### H.1  STATISTICS OF LLM GAME-SOLVING ACCURACY EVALUATION

Table 4 summarizes the results of LLM agents playing long-term mathematical baselines by `o3-2025-04-16`. Here we present experiments conducted under the unbounded value setting, where the maximum achievable payoff for a player is $2/3$. The results show that under fixed roles, the proposer can almost fully capture the entire surplus, implying that the responder receives close to zero. In contrast, the alternating roles condition leads to more balanced outcomes. More specifically, in 10 out of 12 experiments, the final proposer payoff is around 0.33, with two outliers at 0.66 that raise the overall average. Detailed experimental data can be found in the supplementary material. These results suggest that the outcomes of long-term persuasion are similar to those of long-term bargaining, and that alternating offers indeed lead to fairer outcomes, consistent with theoretical predictions.

### H.2  SAMPLE RESULTS

The complete results of all experiments are provided in the supplementary material. Here we present a subset of the experimental results for the model `o3-2025-04-16`.

Table 5: Summary statistics of 87 experiments by `o3-2025-04-16`.

| Experiment | Consensus Rate | Deal Timestep | Final Proposer Payoff |
|---|---|---|---|
| 1 | 1.00 | $1.00 \pm 0.00$ | $1.00 \pm 0.00$ |
| 2 | 0.92 | $1.00 \pm 0.00$ | $0.92 \pm 0.29$ |
| 3 | 0.83 | $1.00 \pm 0.00$ | $0.83 \pm 0.39$ |
| 4 | 1.00 | $1.00 \pm 0.00$ | $1.00 \pm 0.00$ |
| 5 | 0.92 | $1.00 \pm 0.00$ | $0.92 \pm 0.29$ |
| 6 | 0.75 | $1.00 \pm 0.00$ | $0.75 \pm 0.45$ |
| 7 | 1.00 | $1.00 \pm 0.00$ | $0.67 \pm 0.00$ |
| 8 | 1.00 | $1.00 \pm 0.00$ | $0.67 \pm 0.00$ |
| 9 | 0.75 | $1.00 \pm 0.00$ | $0.50 \pm 0.30$ |
| 10 | 1.00 | $1.00 \pm 0.00$ | $0.67 \pm 0.00$ |
| 11 | 0.83 | $1.00 \pm 0.00$ | $0.56 \pm 0.26$ |
| 12 | 0.58 | $1.00 \pm 0.00$ | $0.39 \pm 0.34$ |
| 13 | 1.00 | $1.00 \pm 0.00$ | $1.00 \pm 0.00$ |
| 14 | 1.00 | $1.00 \pm 0.00$ | $1.00 \pm 0.00$ |
| 15 | 0.83 | $1.00 \pm 0.00$ | $0.83 \pm 0.39$ |
| 16 | 1.00 | $1.00 \pm 0.00$ | $1.00 \pm 0.00$ |
| 17 | 1.00 | $1.00 \pm 0.00$ | $1.00 \pm 0.00$ |
| 18 | 0.92 | $1.00 \pm 0.00$ | $0.92 \pm 0.29$ |
| 19 | 1.00 | $1.00 \pm 0.00$ | $0.67 \pm 0.00$ |
| 20 | 1.00 | $1.00 \pm 0.00$ | $0.67 \pm 0.00$ |
| 21 | 0.83 | $1.00 \pm 0.00$ | $0.54 \pm 0.26$ |
| 22 | 1.00 | $1.00 \pm 0.00$ | $0.67 \pm 0.00$ |
| 23 | 1.00 | $1.00 \pm 0.00$ | $0.67 \pm 0.00$ |

| Experiment | Consensus Rate | Deal Timestep | Final Proposer Payoff |
|---|---|---|---|
| 24 | 0.75 | $1.00 \pm 0.00$ | $0.50 \pm 0.30$ |
| 25 | 1.00 | $1.00 \pm 0.00$ | $1.00 \pm 0.00$ |
| 26 | 0.92 | $1.00 \pm 0.00$ | $0.92 \pm 0.29$ |
| 27 | 0.58 | $1.00 \pm 0.00$ | $0.58 \pm 0.51$ |
| 28 | 1.00 | $1.00 \pm 0.00$ | $1.00 \pm 0.00$ |
| 29 | 0.92 | $1.00 \pm 0.00$ | $0.92 \pm 0.29$ |
| 30 | 0.92 | $1.00 \pm 0.00$ | $0.92 \pm 0.29$ |
| 31 | 1.00 | $1.00 \pm 0.00$ | $0.67 \pm 0.00$ |
| 32 | 0.92 | $1.00 \pm 0.00$ | $0.61 \pm 0.19$ |
| 33 | 0.83 | $1.00 \pm 0.00$ | $0.55 \pm 0.26$ |
| 34 | 1.00 | $1.00 \pm 0.00$ | $0.67 \pm 0.00$ |
| 35 | 0.92 | $1.00 \pm 0.00$ | $0.61 \pm 0.19$ |
| 36 | 0.67 | $1.00 \pm 0.00$ | $0.44 \pm 0.33$ |
| 37 | 1.00 | $1.00 \pm 0.00$ | $1.00 \pm 0.00$ |
| 38 | 1.00 | $1.00 \pm 0.00$ | $1.00 \pm 0.00$ |
| 39 | 0.75 | $1.00 \pm 0.00$ | $0.75 \pm 0.45$ |
| 40 | 1.00 | $1.00 \pm 0.00$ | $1.00 \pm 0.00$ |
| 41 | 1.00 | $1.00 \pm 0.00$ | $1.00 \pm 0.00$ |
| 42 | 0.67 | $1.00 \pm 0.00$ | $0.67 \pm 0.49$ |
| 43 | 1.00 | $1.00 \pm 0.00$ | $0.67 \pm 0.00$ |
| 44 | 0.92 | $1.00 \pm 0.00$ | $0.61 \pm 0.19$ |
| 45 | 0.67 | $1.00 \pm 0.00$ | $0.44 \pm 0.33$ |
| 46 | 1.00 | $1.00 \pm 0.00$ | $0.67 \pm 0.00$ |
| 47 | 1.00 | $1.00 \pm 0.00$ | $0.67 \pm 0.00$ |
| 48 | 0.83 | $1.00 \pm 0.00$ | $0.56 \pm 0.26$ |
| 49 | 1.00 | $1.17 \pm 0.58$ | $0.37 \pm 0.07$ |
| 50 | 1.00 | $1.17 \pm 0.39$ | $1.00 \pm 0.00$ |
| 51 | 1.00 | $1.00 \pm 0.00$ | $1.00 \pm 0.00$ |
| 52 | 1.00 | $2.17 \pm 1.53$ | $0.40 \pm 0.12$ |
| 53 | 1.00 | $1.42 \pm 0.67$ | $0.64 \pm 0.10$ |
| 54 | 1.00 | $1.33 \pm 0.65$ | $0.67 \pm 0.00$ |
| 55 | 1.00 | $1.00 \pm 0.00$ | $0.37 \pm 0.07$ |
| 56 | 1.00 | $1.00 \pm 0.00$ | $1.00 \pm 0.00$ |
| 57 | 1.00 | $1.00 \pm 0.00$ | $1.00 \pm 0.00$ |
| 58 | 1.00 | $1.42 \pm 0.51$ | $0.35 \pm 0.00$ |
| 59 | 1.00 | $1.17 \pm 0.39$ | $0.67 \pm 0.00$ |
| 60 | 1.00 | $1.08 \pm 0.29$ | $0.66 \pm 0.00$ |
| 61 | 1.00 | $1.00 \pm 0.00$ | $0.40 \pm 0.09$ |
| 62 | 1.00 | $1.08 \pm 0.29$ | $1.00 \pm 0.00$ |
| 63 | 1.00 | $1.25 \pm 0.45$ | $0.98 \pm 0.03$ |
| 64 | 1.00 | $1.67 \pm 0.89$ | $0.35 \pm 0.00$ |
| 65 | 1.00 | $1.25 \pm 0.45$ | $0.67 \pm 0.00$ |
| 66 | 1.00 | $1.08 \pm 0.29$ | $0.67 \pm 0.00$ |
| 67 | 0.92 | $1.08 \pm 0.29$ | $0.38 \pm 0.15$ |
| 68 | 1.00 | $1.17 \pm 0.39$ | $1.00 \pm 0.00$ |
| 69 | 1.00 | $1.00 \pm 0.00$ | $1.00 \pm 0.00$ |
| 70 | 0.75 | $2.25 \pm 1.60$ | $0.26 \pm 0.16$ |
| 71 | 1.00 | $1.25 \pm 0.45$ | $0.67 \pm 0.00$ |
| 72 | 1.00 | $1.33 \pm 0.65$ | $0.66 \pm 0.03$ |
| 73 | 0.92 | $1.00 \pm 0.00$ | $0.39 \pm 0.19$ |
| 74 | 1.00 | $1.00 \pm 0.00$ | $0.52 \pm 0.17$ |
| 75 | 0.92 | $1.00 \pm 0.00$ | $0.61 \pm 0.19$ |
| 76 | 1.00 | $1.00 \pm 0.00$ | $0.44 \pm 0.16$ |
| 77 | 0.92 | $1.00 \pm 0.00$ | $0.39 \pm 0.19$ |
| 78 | 0.83 | $1.00 \pm 0.00$ | $0.55 \pm 0.26$ |
| 79 | 1.00 | $1.00 \pm 0.00$ | $0.43 \pm 0.15$ |
| 80 | 1.00 | $1.00 \pm 0.00$ | $0.53 \pm 0.17$ |

| Experiment | Consensus Rate | Deal Timestep | Final Proposer Payoff |
|:---:|:---:|:---:|:---:|
| 81 | 0.83 | $1.00 \pm 0.00$ | $0.55 \pm 0.26$ |
| 82 | 1.00 | $1.25 \pm 0.45$ | $0.49 \pm 0.17$ |
| 83 | 1.00 | $1.08 \pm 0.29$ | $0.66 \pm 0.00$ |
| 84 | 1.00 | $1.00 \pm 0.00$ | $0.47 \pm 0.17$ |
| 85 | 1.00 | $1.08 \pm 0.29$ | $0.64 \pm 0.06$ |
| 86 | 1.00 | $1.00 \pm 0.00$ | $0.36 \pm 0.10$ |
| 87 | 1.00 | $1.00 \pm 0.00$ | $0.66 \pm 0.00$ |

# I  LIMITATIONS AND BROADER IMPACTS

## I.1  LIMITATIONS

According to the NeurIPS checklist guidelines, we discuss limitations along the following dimensions: assumptions, claim scope, performance influence factors, and computational efficiency.

**Assumptions.**  One of the main advantages of our work is that our model weaken the assumptions commonly adopted in the community, as discussed in Appendix A.3. We additionally specify an assumption that the receiver can observe the sender's reward function, which was not stated in previous work. Nevertheless, this is a relatively weak assumption because the sender is already able to observe the receiver's reward function, which makes the added assumption symmetric.

**Claim scope.**  Our claim mainly concerns a theoretical reformulation, and the experiments serve a validating role. Each task in the experiments was run 12 times due to the high cost of API usage. However, LLMs tend to give consistent responses to the same problem in most cases.

**Performance influence factors.**  Our experiments rely on the capabilities of LLMs. To mitigate this dependency, we validate the models using tasks with well-established solutions.

**Computational efficiency.**  In the worst case, the players fail to reach an agreement. The number of steps is determined solely by the stop time of the bargaining phase, which follows a geometric distribution. The resulting complexity is $O(1/p)$, where $p$ is the probability of stopping at each step.

## I.2  BROADER IMPACTS

One of the contributions of this work is to improve the generality and applicability of Bayesian persuasion. We predict that when Bayesian persuasion is applied in real-world scenarios, it is likely to trigger bargaining behavior from the receiver if the receiver is aware of the sender's reward function. Through our proposed information bargaining framework, it is possible to formulate solution concepts with desirable properties, including fairness and Pareto efficiency. Therefore, the overall impact is expected to be socially beneficial.

