# OpenReview forum: "Information Bargaining: Bilateral Commitment in Bayesian Persuasion"
_ICLR.cc/2026/Conference — Submitted to ICLR 2026_

### Official Review · Reviewer_ibGS · 2025-10-27

**Soundness:** 1
**Presentation:** 2
**Contribution:** 1
**Rating:** 2
**Confidence:** 4

**Summary:**

The paper proposes a framework that modifies the Bayesian persuasion framework by adding elements of bargaining theory. The authors argue that some form of “long-term” Bayesian persuasion interactions can be decomposed into two stages: a bargaining stage and a realization stage (i.e. executing the strategies resulting from bargaining). The authors argue that this decomposition separates the sender’s “informational advantage” from their “first-mover advantage”.  Finally, some experiments using LLMs as players in the interaction are presented.

**Strengths:**

The general idea of introducing a “bargaining phase” to model bilateral commitment in information exchange is interesting and could in principle broaden the scope of standard persuasion models.

**Weaknesses:**

The paper’s objectives and main contributions are largely unclear. I find neither the proposed model nor the empirical results convincing in their current form.

Starting with the model. The paper augments the standard Bayesian persuasion framework with an additional bargaining phase. However, the motivation for this modification is not discussed. The discussion around lines 52 and 58–60 does not provide a solid justification, nor does it cite evidence for the “truthfulness discrepancy” it mentions. Since this is a substantial departure from the standard model, a more convincing argument is needed to explain why the extension is necessary. At present the modification may appear ad hoc, with the byproduct of having a setup for LLM-based experiments.

The proposed framework also introduces strong assumptions that are not discussed or justified. In particular, the assumption of complete mutual knowledge of preferences (e.g., the receiver knowing the sender’s utility function) is quite restrictive and not standard in Bayesian persuasion. The paper should discuss why this is reasonable, since most of the subsequent discussion is based on this assumption.

In terms of exposition, the model description is vague and not formal. The paper repeatedly refers to “long-term persuasion” but never clearly defines what this means, nor does it relate the setup to existing formulations of online Bayesian persuasion (in case “long-term” refers to this). Similarly, the repeated claim that the proposed decomposition “does not change optimality or equilibrium” is stated without formal proof or explanation.

The theoretical insights are quite weak in their current form. Section 3 mainly states straightforward implications of giving both players full information about expected payoffs. Lemma 4.3 is a simple observation once the assumptions are granted and does not represent a substantive theoretical contribution.

The empirical validation adds little to the argument. I don’t think the current evaluation provides meaningful evidence for the claims (or maybe it does, but it’s not clear from the current exposition). It remains unclear what can be learned about persuasion or bargaining from these experiments, and why LLM-based simulation should matter for the argument being made.

Clarity

Section 2.2 is very vague and does not clarify the notion of bargaining game unless the reader is already familiar with it. It would be good to give an example here.

Several concepts in the paper are not defined (e.g., visibility sets at line 150, extensive form games at line 214, babbling equilibrium at line 271, shadow of the future)

Example at line 273 is not clear.

**Questions:**

none

---

### Official Review · Reviewer_RRXQ · 2025-11-01

**Soundness:** 3
**Presentation:** 4
**Contribution:** 2
**Rating:** 2
**Confidence:** 4

**Summary:**

This paper studies a variant of the Bayesian persuasion problem in which the receiver has perfect knowledge of the sender’s utility function and behaves strategically under different signaling schemes. The authors show that this variant can be reduced to a bargaining game. In particular, the receiver can use a threat to induce the sender to adopt a signaling scheme that is more favorable to the receiver. The authors also conduct experiments with LLMs.

**Strengths:**

The paper is clearly written and easy to follow. The variant of Bayesian persuasion studied in this paper is conceptually interesting. The idea that the receiver may strategically threaten the sender by refusing to follow the sender’s signaling scheme is natural and may capture a realistic strategic interaction.

**Weaknesses:**

The Bayesian persuasion variant proposed in this paper relies heavily on the assumption that the sender's utility function is known to the receiver. However, this assumption is only briefly mentioned (line 153) and lacks sufficient justification. The paper could benefit from a more detailed discussion of when and why this assumption might hold in practice.

The paper refers to the Long-Term Bayesian Persuasion problem, which is known to be NP-hard, and decomposes it into a bargaining stage and a realization stage. However, in the literature, the prior on states in Long-Term Bayesian persuasion typically evolves over rounds. In contrast, Procedure 2 in the paper assumes a fixed prior across rounds, which turns the problem into a repeated game, a much simpler setting. The difference between these two formulations is substantial. It would be valuable to discuss how the results might change if the prior were updated dynamically.

A significant portion of the paper is devoted to reviewing Bayesian persuasion and bargaining games. While some background is necessary, the exposition could be shortened to make room for deeper analysis or additional results. The theoretical contribution currently centers on a single result and lacks theoretic depth. The experiments are also not very convincing as outputs of LLMs may vary depending on the prompts. The results are also not surprising. Simply proposing a new setting with a new solution concept and using LLMs for experiments do not guarantee acceptance to top-tier conferences like ICLR.

**Questions:**

Please see my comments above.

---

### Official Review · Reviewer_Yaqh · 2025-11-01

**Soundness:** 2
**Presentation:** 2
**Contribution:** 3
**Rating:** 4
**Confidence:** 2

**Summary:**

This paper introduces the bargaining perspective to Bayesian persuasion (BP).  The paper defines a BP game where, unlike the classical BP where only the sender has commitment power (commit to signaling schemes), now the receiver can also commit (to their mapping from signals to actions).  The paper seems to show that the BP problem with joint commitment can be reduced to a long-term Bayesian persuasion problem with a bargaining phase (subject to some criticism I mentioned in Weakness 1.2.2).  Then, the paper conducts experiments using LLM to play the bargaining game and BP game, validating the hypothesis that these two games have identical equilibrium outcomes.

**Strengths:**

(S1) This paper introduces an interesting new perspective to the study of long-term Bayesian persuasion problem, namely, the bargaining perspective: the sender and receiver can bargain over the signaling scheme before implementing it.

(S2) LLM experiments demonstrate that the bargaining solution matches the classical Bayesian persuasion solution, meaning that the critical commitment assumption in classical BP can arise from bargaining.  This is an interesting observation.

**Weaknesses:**

(1) The paper's first contribution is claimed to be "We show that long-term persuasion problems can be decomposed into a bargaining stage and a realization stage without affecting optimality or equilibria".  I don't think this claim is supported well mathematically.

(1.1) First, the "long-term persuasion problem" is never mathematically defined. Section 2 defines one-shot Bayesian persuasion and cheap talk games.  And Section 4 directly introduces a model of "long-term persuasion with two stages (bargaining and realization)".  I thought the authors wanted to show that "long-term persuasion problem __(without bargaining)__ can be decomposed into a bargaining stage and a realization stage". However, "long-term persuasion problem" (without bargaining) is never defined, in my understanding.

(1.2) Then, I tried to read Lemma 4.3 to understand what this claim truly means.  The lemma says that a Bayesian persuasion task is reducible to a bargaining game, where "reducible" means any solution concept in bargaining game can be turned into a solution concept in Bayesian persuasion game.  There are two issues with this lemma:

  (1.2.1) The Bayesian persuasion game in this lemma is not the traditional Bayesian persuasion game where only the sender has commitment power; instead, it is a game with joint commitment where the receiver can also commit.  So, this lemma is proving "BP with joint commitment can be reduced to bargaining game", instead of "classical BP can be reduced to bargaining game".  BP with joint commitment is unconventional and lacks motivation: why does the receiver have commitment power?  I think the authors might want to do the reduction in the opposite direction: showing that the bargaining game can be reduced to BP with joint commitment.  Given that reduction, we can then use BP with joint commitment to study the bargaining game (which is a more natural problem than BP with joint commitment in my opinion).

  (1.2.2) The bargaining game constructed in this reduction (in Appendix E) is trivial: it only contains the joint payoff profile $(R^i, R^j)$ that is induced by the sender's signaling scheme and the receiver's response in the solution of the Bayesian persuasion problem (with joint commitment).  Because the bargaining game only contains this profile, then clearly any solution concept of the bargaining game will pick this solution, making the argument trivial.  Usually, a bargaining game should contain a set $\mathbb Y$ of multiple feasible agreements, and a solution concept picks an agreement from $\mathbb Y$.  The bargaining game is meaningful only when $\mathbb Y$ contains multiple agreements, but the bargaining game in this lemma only has a single agreement.  The correct argument should, in my opinion, consider the bargaining game where the feasible set of agreements consists of the payoff profiles associated with all feasible pairs of signaling scheme and receiver response, and argue that the bargaining solution concept picks the agreement that is equal to the BP solution.



(2) The paper's second contribution is "We clarify two advantages that have been conflated in Bayesian persuasion, namely the sender's information advantage and the first-mover advantage".  It's unclear to me how these two advantages are "clarified" mathematically.  Do you mean that, when one advantage is removed, the solution of Bayesian persuasion will change?  And the sender's payoff will decrease when the advantage is removed?  These two advantages are already known in previous literature. The long-term Bayesian persuasion problem considered in this paper seems to still allow the sender to have these two advantages.  I don't know if this paper provides any new observations about these two advantages.

**Questions:**

## Questions for the authors

As written in the Weaknesses, I am confused by many parts of this paper.  I would appreciate it if the authors could provide some clarifications.



## Suggestions

Typos:

* Definition 2.3: $\varphi(\sigma=\sigma | s)$ and $\varphi(\sigma|s)$ should be $\varphi(\sigma=a | s)$
* Line 215: "at uniformly random" -> "uniformly at random"
* Line 240: "$\sum_{s' | \sigma} \mu(s'|a) \sum_a \pi(a|s')$" should be "$\sum_{s'} \mu(s'|\sigma) \sum_a \pi(a|\sigma) = 1 \cdot \sum_a \pi(a|\sigma)$", I think

---

### Official Review · Reviewer_LB38 · 2025-11-03

**Soundness:** 3
**Presentation:** 3
**Contribution:** 2
**Rating:** 4
**Confidence:** 4

**Summary:**

The paper introduces information bargaining, a two-stage view of long-term persuasion: a bargaining stage where sender and receiver commit to a signaling scheme and an action rule, followed by a realization stage where signals and actions are executed. The key claims are: the decomposition leaves optimality and equilibria unchanged; the sender’s power splits into informational advantage and first-mover advantage; and Bayesian persuasion can be reduced in polynomial time to a cooperative bargaining problem, enabling use of bargaining solution concepts such as the Nash bargaining solution. The paper formalizes a disagreement point via prior-based “babbling,” and proposes joint commitment as a fixed point of simultaneous policy updates.

To validate predictions, the authors use LLMs as equilibrium solvers. They first screen models on games with known solutions using Pearson correlation between model payoffs and ground truth, then run 87 bargaining and persuasion variants and check correlation with their hypotheses. Only two “reasoning” models pass the screen; reported correlations are high. Prompts force each agent to role-play a “self-interested, rational” player and to output JSON decisions. The authors state that these experiments are proof-of-concept and that the contribution is primarily theoretical.

**Strengths:**

The theoretical framing is clear and potentially useful. The bargaining–realization split exposes the sender’s first-mover advantage and clarifies fairness considerations. The reduction from persuasion to bargaining gives a concrete path to bring cooperative bargaining solution concepts to information design. Definitions of disagreement points and joint commitment are explicit, with timing procedures that make the modeling choices inspectable.

**Weaknesses:**

The empirical evaluation relies on LLM rationality. The experiments assume that an LLM instantiated by a prompt is a rational best-response agent in the game-theoretic sense. This assumption is very challengable. The screening metric is payoff correlation, which can be high even if strategies are not equilibria, obedience fails, or off-path contingencies are wrong. Prompt structure, temperature, top-k, and other parameters including the random seed can change an LLM's output, which, in my view, renders the results it yields a little sketchy. The use of chain-of-thought is motivated as transparency, which is also rather debatable -- CoT has nothing to do with the internals of an LLM; it's just a multistaged generation process.
Also, there is no comparison to exact or approximate solvers on small games or to human subjects with real incentives.

**Questions:**

Re: the use of LLMs, can you report best-response regret for each agent under the learned signaling scheme and action rule? That is, compute the gain from a unilateral deviation. This would test rationality directly rather than via payoff correlation.

Can you show some measure of sensitivity to prompt structure, temperature, seed, and so on, also (and in particular) reporting standard deviations?

---

### Meta-Review · Area_Chair_TaZL · 2025-12-20

**Summary:**

The reviewers pointed out several weaknesses and shared common concerns, including (a) a lack of sufficient empirical evidence supporting the claims as it relies primarily on LLM-based simulations; (b) issues with mathematical formulation, definitions, and assumptions that are central to the main results.

**Reviewer Concerns:**

There was no rebuttal for this paper, and I believe the reviewers’ concerns raised in the original reviews are still outstanding.

**Reviewer Scores:**

There was no rebuttal for this paper, and I don’t think the reviewers would have changed their scores without a rebuttal.

---

### Decision · Program_Chairs · 2026-01-26

Reject